



**1 Airborne Lidar Observations of Wind, Water Vapor, and Aerosol Profiles During The**
**2 NASA Aeolus Cal/Val Test Flight Campaign**

Kristopher M. Bedka[1], Amin R. Nehrir[1], Michael Kavaya[1], Rory Barton-Grimley[1], Mark
Beaubien[6], Brian Carroll[5], James Collins[2], John Cooney[5], G. David Emmitt[3], Steven Greco[3],
Susan Kooi[2], Tsengdar Lee[4], Zhaoyan Liu[1], Sharon Rodier[2], Gail Skofronick-Jackson[4]
[1]NASA Langley Research Center, Hampton, VA
[2]Science Systems and Applications, Inc., Hampton, VA
[3]Simpson Weather Associates, Charlottesville, VA
[4]NASA Headquarters, Washington D.C.
[5]NASA Postdoctoral Fellowship Program, Universities Space Research Association at NASA Langley Research
Center, Hampton, VA
[6]Yankee Environmental Systems, Inc., Turners Falls, MA
*Correspondence to:* Kristopher M. Bedka (kristopher.m.bedka@nasa.gov)

**18 ABSTRACT**

Lidars are uniquely capable of collecting high precision and high spatio-temporal resolution observations that have
been used for atmospheric process studies from the ground, aircraft, and space for many years. The Aeolus mission,
the first space-borne Doppler wind lidar, was developed by the European Space Agency and launched in August 2018.
Its novel Atmospheric LAser Doppler INstrument (ALADIN) observes profiles of the component of the wind vector
and aerosol/cloud optical properties along the instrument's line-of-sight (LOS) direction on a global scale. Two
airborne lidar systems have been developed at NASA Langley Research Center in recent years that collect
measurements in support of several NASA Earth Science Division focus areas. The coherent Doppler Aerosol WiNd
(DAWN) lidar measures vertical profiles of LOS velocity along selected azimuth angles that are combined to derive
profiles of horizontal wind speed and direction. The High Altitude Lidar Observatory (HALO) measures high
resolution profiles of atmospheric water vapor (WV), and aerosol and cloud optical properties. Because there are
limitations in terms of spatial and vertical detail and measurement precision that can be accomplished from space,
airborne remote sensing observations like those from DAWN and HALO are required to fill these observational gaps
as well as to calibrate and validate space-borne measurements.
Over a two-week period in April 2019 during their Aeolus Cal/Val Test Flight campaign, NASA conducted five
research flights over the Eastern Pacific Ocean with the DC-8 aircraft. The purpose was to demonstrate: 1) DAWN
and HALO measurement capabilities across a range of atmospheric conditions, 2) Aeolus Cal/Val flight strategies and
comparisons of DAWN and HALO measurements with Aeolus to gain an initial perspective of Aeolus performance,
and 3) how atmospheric dynamic processes can be resolved and better understood through simultaneous observations



of wind, WV, and aerosol profile observations, coupled with numerical model and other remote sensing observations.
This paper provides a brief description of the DAWN and HALO instruments, discusses the synergistic observations
collected across a wide range of atmospheric conditions sampled during the DC-8 flights, and gives a brief summary
of the validation of DAWN, HALO, and Aeolus observations and comparisons.



## 1. Introduction

The Aeolus mission, the first-ever space-borne Doppler wind lidar (DWL), was developed by the European Space Agency (ESA) and launched in August 2018. Aeolus has a sun-synchronous orbit at 320 km altitude (Kanitz et al. 2019), and carries a single payload, the Atmospheric LAser Doppler INstrument (ALADIN). ALADIN observes profiles of the component of the wind vector along the instrument's line-of-sight (LOS) direction, on a global scale from the ground up to 30 km in the stratosphere (ESA 2016; Stoffelen et al. 2005; Reitebuch 2012; Kanitz et al. 2019). Aerosol optical properties are retrieved from ALADIN measurements employing an interferometric approach similar to the High Spectral Resolution Lidar (HSRL) technique (Shipley et al. 1983; Flamant et al. 2008). Aeolus observes horizontal wind speed and aerosol profiles along a single LOS. The Aeolus mission serves as both a technology demonstration as well as validation of predicted impacts of global wind profile observations on weather forecasting and atmospheric research. There is currently a robust international effort to conduct intensive Aeolus calibration and validation (Cal/Val) using ground and suborbital remote and in situ sensors as well as comparison against numerical model background fields (ESA 2019; Witschas et al. 2020; Lux et al. 2020; Baars et al. 2020; Martin et al. 2020; Khaykin et al. 2020). NASA's longstanding heritage in development and deployment of airborne DWL technologies, coupled with its interests in utilizing space-borne wind and aerosol observations for Earth system science process studies and weather prediction, served as the primary motivating factors for the agency to contribute to the Aeolus Cal/Val effort.

The 2017 Decadal Survey for Earth Science and Applications from Space (ESAS 2017) identified a set of key Earth science and applications questions to be addressed by the research community over the next decade. Two of these questions were related to 1) a better understanding planetary boundary layer (PBL) processes and air-surface fluxes and 2) an understanding of why clouds, convection, heavy precipitation, occur when and where they do. Measurements required to address these questions include aerosol vertical profiles and properties, PBL height, cloud type, depth and hydrometeor composition, temperature, water vapor and wind profiles, as well as many other geophysical variables. Observations of these variables are critical to numerical weather prediction models and reanalyses that have informed our understanding of the Earth's weather and climate system over the last 40+ years (Stith et al. 2018). A number of active and passive space-borne remote sensing systems collect such observations, however, they often lack horizontal and vertical resolution, spatial coverage, temporal frequency, and/or precision to enable detailed process studies and advance our understanding. ESAS (2017) recommends a future Earth System

Explorer class mission focusing on atmospheric wind measurement to address gaps in the current space-borne wind
observing network. However, there are limitations as to how processes can be observed from space, thus ESAS (2017)
also recommends airborne and ground based in-situ and remote sensing to fill observational gaps.
Lidars are uniquely capable of collecting high precision and high spatio-temporal resolution observations
that have been used for atmospheric process studies from the ground, aircraft, and space for many years. Two airborne
lidar systems have been developed at NASA Langley Research Center (LaRC) in recent years that collect
measurements in support of the NASA Science Mission Directorate Earth Science Division Weather and Atmospheric
Dynamics, Carbon Cycle, and Atmospheric Composition Science Focus Areas. Initially developed in the late 2000's,
the Doppler Aerosol WiNd (DAWN, Kavaya et al. 2014) lidar measures vertical profiles of LOS velocity that are
combined to derive horizontal wind speed and direction using the coherent detection method (see Henderson et al.
2005 and references therein). More recently in 2015, the High Altitude Lidar Observatory (HALO), was developed to
measure high resolution profiles of atmospheric water vapor (WV), aerosol/cloud optical properties. HALO also
provides the option to substitute WV profile observations with a total column and mixed layer methane observation
(Nehrir et al. 2018). The distribution of atmospheric WV and its coupling to circulation is a common focus across
several of the World Climate Research Program (WCRP) Grand Challenges and progress requires improved
understanding of processes based on "observations of WV, winds and clouds, especially in the lower troposphere, and
at higher vertical resolution than is available from current sensors" (Asrar et al. 2015; Wulfmeyer et al. 2015; Nehrir
et al. 2017; Stevens et al. 2017). Simultaneous, high spatio-temporal resolution (< 0.5 km vertical, 1-10 km spatial)
wind, WV, and aerosol observations from DAWN and HALO serve as an ideal remote sensing payload for supporting
a breadth of airborne science campaigns to address the key process-oriented science questions posed by the 2017
Decadal Survey and WCRP, as well as for satellite Cal/Val activities such as for the Aeolus mission.
In April 2019, NASA conducted an Aeolus Cal/Val Test Flight campaign to demonstrate: 1) DAWN Doppler
wind lidar and new HALO HRSL aerosol and WV DIAL observational capabilities across a range of atmospheric
conditions, 2) flight strategies for Aeolus Cal/Val and comparisons with Aeolus to gain an initial perspective of Aeolus
performance, in preparation for future international Aeolus Cal/Val airborne campaigns, and 3) how atmospheric
dynamic processes can be resolved and better understood through simultaneous observations of wind, WV, and aerosol
profile observations, coupled with numerical model and other remote sensing observations. Five NASA DC-8 aircraft
flights were conducted over a period of two weeks over the Eastern Pacific and Southwest U.S., based out of NASA


Armstrong Flight Research Center in Palmdale, California and Kona, Hawaii. Dropsondes were used to validate the
DAWN Aeolus wind observations and HALO WV observations. The LaRC Diode Laser Hygrometer (Diskin et al.
2002) was also installed on the DC-8 and provided several in-situ WV validation profiles for both the HALO and
dropsonde measurements.

This paper provides a brief description of the DAWN and HALO instruments, discusses the synergistic

observations collected across a wide range of atmospheric conditions sampled during several DC-8 flights, and a
summary of the validation of DAWN, HALO, and Aeolus observations and comparisons.

**2. Instrument Overview**
**2.1 DAWN**
DAWN, a pulsed 2-micron coherent-detection doppler wind lidar (DWL), was initially developed in the 2000's at
NASA LaRC as an airborne instrument simulator to demonstrate technologies that would be required for a future
space-borne doppler wind lidar mission as well as to support airborne process studies and satellite Cal/Val activities.
DAWN is one of several airborne DWLs operated by the international community, such as those described by Wang
et al. (2012), Witschas et al. (2017); Bucci et al. (2018), Lux et al. (2018), Marksteiner et al. (2018), Tucker et al.
(2018), and Zhang et al. (2018).

An overview of the DAWN system architecture is described by Kavaya et al. (2014) and Greco et al. (2020),

but a brief summary is provided here as background. During the 2019 Aeolus Cal/Val campaign, DAWN operated
with a 10 Hz pulse repetition rate and 180 ns pulse duration, generating ~100 mJ per pulse. It originally generated 250
mJ per pulse using a crystal amplifier for the GRIP and PolarWinds I and II campaigns described below. However,
this component failed and the space it occupied was used for critical beam shaping optics that increased signal-to-
noise ratio (SNR) significantly more than if the amplifier had been replaced. DAWN utilizes a 30° deflecting wedge
scanner at the output of the system beam expander to enable vertical profiling of horizontal wind vectors. DAWN can
scan at user-specified azimuth angles (commonly referred to as "looks") with a user-specified number of laser pulse
averages per LOS wind profile. Generally, a greater number of azimuths and pulses/azimuth improves vertical
coverage of successful wind retrievals and cloud cover penetration, respectively, both at the expense of horizontal
distance between profiles. DAWN also has the ability to stare at one azimuth angle to retrieve wind speed along the
LOS which is analogous to Aeolus observations.  Table 1 provides a summary of DAWN operating modes during the



campaign.  The nominal operating mode is 5 azimuths (45°, 22.5°, 0°, -22.5°, -45°), with 0° oriented forward along
flight track, and 20 pulses/azimuth, providing wind profiles every 4-5 km assuming nominal DC-8 cruise speeds of
225-250 m/s and ~2 seconds to move the scanner to a new azimuth.

DAWN wind retrievals are based on methods developed within the coherent DWL community, described by

Kavaya et al. (2014) and Greco et al. (2020). The range-resolved retrieval is applied to each range gate spaced by 128
samples corresponding to an along LOS range of ~ 38 m at a 500 MHz sampling rate, which projects as 33 m in the
vertical with a 30° off-nadir angle. For each retrieval, a range gate of 256 samples is zero padded to 1024 samples in
the FFT spectral analysis to determine a Doppler shift in the received lidar signal. A 256 sample range gate computes
to be 76 m along the LOS and 66 m in the vertical. The 180 ns DAWN pulse full width at half maximum (FWHM)
yields a folded 27 m convoluted return along a LOS. With the 30° off nadir angle, that corresponds to a ~23 m vertical
return volume. In addition, the electronic bandwidth of 250 MHz yields an along LOS range of 0.6 m and a vertical
range of 0.52 m. Therefore the current DAWN wind measurement has a minimum vertical range resolution of ~90 m,
but wind retrievals in each profile are spaced by 33 m in the vertical thus there is some correlation between adjacent
vertical levels. FFT periodograms are averaged across the number of pulses per azimuth, with corrections employed
to account for slight shot-to-shot shifts in transmitted laser frequency across the number of pulses. This shot
accumulation improves SNR and thus permits wind measurements to have a higher success rate in lower
concentrations of aerosols. The frequency shift is combined with aircraft airspeed and attitude information from the
DC-8 INS/GPS system to derive a wind speed range profile along the LOS. Winds from multiple LOS are combined
within a solver of a linear system of equations to derive a wind vector. In practice, the vertical wind is assumed zero
to reduce the degrees of freedom.

If a successful wind retrieval could not be derived at the highest vertical resolution (a slant range of 76.7

meters) due to low aerosol concentration, data is integrated in the vertical over increasingly deep layers of 153.4,
306.8, 613.6, 1227.3 meters along each line of sight until a sufficient signal magnitude at least 1.5 times the standard
deviation of the periodogram's noise floor power is achieved.  This approach is similar to the Adaptive Sample
Integration Algorithm (ASIA) described by Greco et al. (2020). However, unlike Greco et al. (2020) where a wind is
retrieved from samples of each LOS at the same slant range possibly corresponding to different altitudes especially
during the aircraft turns and ascent/descent, a DAWN wind retrieval is performed using samples of each LOS at the
same altitude. This can improve wind retrievals during aircraft maneuvers. Figures 3b and 10d shows that most winds



are retrieved via multi-pulse integration at a single range bin (purple), analogous to the "base" retrievals described by
Greco et al. (2020), but lower aerosol backscatter at middle levels (~2-5 km altitude) of the profile in the free
troposphere required more vertical integration to achieve sufficient signal.

DAWN has been used for a variety of NASA studies over the last decade. DAWN first flew on the DC-8

during the 2010 NASA Genesis and Rapid Intensification Processes (GRIP) campaign (Braun et al. 2013; Kavaya et
al. 2014). DAWN was also operated from the ground to explore opportunities for wind energy applications offshore
of Virginia (Koch et al. 2012). Additionally, DAWN participated in two flight campaigns, PolarWinds I and II, in
2014 with the NASA UC-12B, and 2015 with the DC-8, respectively. PolarWinds II (Marksteiner et al. 2018) involved
collaboration with the European Space Agency and the Deutsches Zentrum für Luft- und Raumfahrt (DLR). The
NASA DC-8 and the DLR Falcon-20 exercised coordinated flight strategies during PolarWinds II that helped to inform
the 2019 Cal/Val campaign strategy as well as future campaigns. During PolarWinds II, DAWN provided the first
airborne DWL observations of a mesoscale barrier jet, driven by the interaction of synoptic scale wind with the steep
and complex topography of Greenland (DuVivier et al. 2017). DAWN also flew aboard the DC-8 during the 2017
Convective Processes Experiment (CPEX), a campaign which sought to better understand convective cloud dynamics,
downdrafts, cold pools and thermodynamics during initiation, growth, and dissipation, as well as to improve model
representation of convective and boundary layer processes through assimilation of DAWN and other remote sensing
and in-situ observations (NASA 2017). DAWN wind profiles agreed very well with 169 dropsonde profiles in a
variety of cloud, wind, and aerosol conditions, with less than 0.2 m/s bias (or "accuracy") and 1.6 m/s root-mean-
squared difference (RMSD, or "precision") for wind components (Greco et al. 2020). DAWN wind observations were
also well-correlated with flight-level winds measured in-situ by the DC-8 and near-surface wind measured by buoys.
CPEX DAWN data were assimilated into a mesoscale model to improve simulations of a mesoscale convective system
and tropical storm (Cui et al. 2020) and used to demonstrate how airborne radar and Doppler wind lidar data can be
used together to study convective processes (Turk et al. 2020).

A 2-micron coherent detection Doppler wind lidar has been used by DLR for almost 20 years within a variety

of airborne science campaigns. Witschas et al. (2020) describes that detailed comparisons between the DLR lidar and
sonde demonstrate a bias of this system to be < 0.10 m/s and a scaled median absolute deviation (approximately the
same as RMSD due to minimal bias) between 0.9 and 1.5 m/s. This level of performance was deemed suitable for
Aeolus Cal/Val during the DLR WindVal III and Aeolus Validation Through Airborne Lidars in Europe (AVATARE)



campaigns. Though the DLR pulse energy, pulse length, and spatial sampling interval differs from DAWN, the
DAWN has provided similar performance to the DLR system in previous campaigns, and therefore is also a useful
benchmark for validating Aeolus wind profiles.
Throughout the 2019 Aeolus campaign, the DAWN INS/GPS unit operated unreliably. This unit was
designed to collect data at 10 Hz that could be synced with each DAWN laser pulse to remove aircraft speed and
attitude effects from the returned atmospheric signal for retrieval of Doppler shift caused by aerosol motions, as
described by Greco et al. (2020). The DC-8 1 Hz INS/GPS data was interpolated to the time of each DAWN pulse
and used in place of the DAWN 10 Hz unit. Based on periods where there were reliable 10 Hz INS/GPS data, we
found that use of 1 Hz data does add some variance to the DAWN wind retrieval, ranging from ~0.1-0.3 m/s and 0.5°-
2.5° degrees in wind speed and direction, respectively based on sensitivity analyses. Attempts to address issues with
the problematic INS/GPS unit resulted in periods generally less than 10 min of DAWN lidar downtime at various
times during the flights.

**2.2 HALO**
NASA Langley Research Center developed HALO to address the observational needs of NASA Earth Science
Division Weather and Atmospheric Dynamics, Carbon Cycle, and Atmospheric Composition Science Focus Areas.
HALO is a modular and multi-function airborne lidar developed to measure atmospheric $H_2O$ and $CH_4$ mixing ratios
and aerosol, cloud, and ocean optical properties using the differential absorption lidar (DIAL, Measures 1984, Nehrir
et al. 2017) and high spectral resolution lidar (HSRL, Hair et al. 2008) techniques, respectively. HALO was designed
as a compact replacement for the LASE WV DIAL (Browell et al. 1997) with improved and substantial additional
capabilities (Nehrir et al. 2017-19). Furthermore, HALO was designed as an airborne simulator for future space-borne
greenhouse gas DIAL missions called for by ESAS (2017) and also serves as testbed for risk reduction of key
technologies required to enable those future space-borne missions. To respond to a wide range of airborne process
studies, HALO can be rapidly reconfigured to provide either, $H_2O$ DIAL/HSRL, $CH_4$ DIAL/HSRL, or $CH_4$ DIAL/$H_2O$
DIAL measurements using three different modular laser transmitters and a single multi-channel and multi-wavelength
receiver. Though HALO has successfully flown in several field campaigns in the $CH_4$ DIAL/HSRL configuration, the
2019 Aeolus Cal/Val campaign was the maiden deployment for the $H_2O$ DIAL/HSRL configuration. Despite serving

Bc:

as the first set of engineering test flights, HALO exceeded all expectations during the Aeolus Cal/Val campaign,
demonstrating the first new airborne WV DIAL capability within NASA in over 25 years.
In the $H_2O$ DIAL/HSRL configuration, HALO employs a 1 KHz pulse repetition frequency injection seeded,
Nd:YAG pumped optical parametric oscillator (OPO) pulsed laser to enable WV profile measurements at 935 nm
using the DIAL technique, as well as the HSRL technique at 532 nm to make independent, unambiguous retrievals of
aerosol extinction and backscatter. It also employs the standard backscatter technique at 1064 nm and is polarization-
sensitive at the 1064/532 nm wavelengths. To enable WV profiling over a large dynamic range throughout the
troposphere, HALO transmits four discrete wavelengths (three wavelength pairs) at 935 nm positioned on and off
varying strength WV absorption lines where each transmitted wavelength pair provides sensitivity to a different part
of the atmosphere. The profiles retrieved from the three transmitted line pairs are spliced together using a weighted
mean where the WV optical depth is used to constrain the upper and lower bounds of the splice region. This WV
sampling approach is similar to that presented by Wirth et al. (2009), however, HALO utilizes a single laser transmitter
to generate all four transmitted wavelengths thereby significantly reducing the overall size, weight and power of the
instrument. An overview paper summarizing the description and performance of HALO and its associated $H_2O$, $CH_4$,
and HSRL measurements is currently in preparation.
HALO data are sampled at 0.5 s temporal and 1.25 m vertical resolution, respectively. Real-time onboard
processing is employed to sum 125 shots at each of the four WV DIAL wavelengths and 500 shots at the 532 nm and
1064 nm wavelengths to reduce the reported data rate to 2 Hz. A high sampling rate of 120 MHz is employed to allow
for accurate $CH_4$ retrievals in the other HALO measurements configurations, as well as future cloud and ocean
profiling. The WV DIAL and 1064 nm backscatter channels have an electrical bandwidth equivalent to 15 m vertical
resolution. The electrical bandwidth for the HSRL channel at 532 nm is matched to the native sampling rate to achieve
1.25 m vertical resolution in the atmosphere. The 532 nm signals are subsequently filtered and binned to 15 m vertical
resolution in post-processing to increase the SNR of the HSRL aerosol retrievals.
The DIAL technique directly measures the molecular number density of water vapor. Conversion to mass or
volume mixing ratio requires knowledge of the dry air number density which is obtained from MERRA-2 reanalysis
fields of atmospheric pressure and temperature (Gelaro et al. 2017) that are interpolated in space and time to the lidar
sampling track and resolution. The HALO water vapor mixing ratio (WVMR) products are averaged over 30-60
seconds horizontally (6-12 km from the DC-8 assuming nominal cruise speed) and 315-585 m vertically to achieve



an absolute precision of better than 10% which are calculated using DIAL error propagation and Poisson photon
statistics. The temporal and vertical averaging can be traded for precision in post processing and optimized for specific
science applications. For the Aeolus Cal/Val campaign, HALO was able to demonstrate a precision of better than
10% with 6 km along track averaging when the water vapor differential absorption optical depth (DAOD) was
optimized through wavelength tuning for the viewing scene. Given that this campaign served as the first check flights
for HALO, optimization of the WV DAOD within the lower troposphere was not achieved for parts of the first flight
as well as the tropical scenes, resulting in loss of precision near the surface where the absorption was too large. For
ease of interpretation across the various flights presented below, all of the HALO WV data are shown at a 12 km fixed
horizontal resolution. To overcome the loss in precision for cases where the near surface DAOD was too large or
SNR was degraded due to cloud attenuation, an adaptive vertical averaging routine is employed where the vertical
resolution is increased from 315 m to 585 m when the uncertainty in the calculated WVMR drops below 10%. A
weighted mean is used to transition between the two different vertical averaging bin sizes. The trades on HALO WV
precision vs vertical and horizontal resolution as well as the adaptive vertical averaging routine will be the subject of
a future paper.

Dropsonde humidity measurements during the Aeolus Cal/Val campaign lacked the vertical resolution and

precision to validate the HALO WVMR retrievals, as is discussed in the following section. Qualitative comparisons,
however, generally showed good agreement in the lower troposphere and into the PBL. Comparisons with the DLH
in-situ open path measurement conducted during a spiral showed excellent agreement and are discussed further in
Section 4.4. A detailed assessment of the HALO WV retrievals is beyond the scope of this paper and will be presented
in a separate manuscript where statistical comparisons against the dropsondes, DLH in-situ measurements and satellite
retrievals of the same geophysical variable will be discussed.

In addition to profiling WVMR throughout the troposphere, total or partial columns of precipitable WV are

obtained by vertical integration of the WVMR profiles. WV profile data above the surface are limited to
approximately the vertical resolution of the retrieval bin width which is required to achieve sufficient on/off extinction
for high precision measurements. WV profiles over the ocean are extended to the surface utilizing the strong surface
echo where a DIAL retrieval is carried out between the last good atmospheric retrieval above the surface and the on/off
absorption from the surface echo. Preliminary results using the ocean surface echo show promise for extending the
WV profile to the surface and compare favorably with in situ observations. Variability in surface height confounds



the surface return retrievals and are not employed over land for this study. Additionally, WV profiles above clouds
are masked with an additional 45 m to avoid cloud edge effects and contamination in the DIAL retrieval.

One of the primary functions of HALO during the Aeolus Cal/Val campaign was to provide aerosol validation

for the ALADIN co-polar aerosol backscatter and extinction products.  The HALO aerosol HSRL and backscatter
retrievals follow the methods presented by Hair et al. (2008).  HALO aerosol backscatter and depolarization products
are averaged 10 s horizontally and aerosol extinction products are averaged 60 s horizontally and 150 m vertically.
The polarization and HSRL gain ratios are calculated as described in Hair et al. (2008). Operational retrievals also
provide mixing ratio of non-spherical-to-spherical backscatter (Sugimoto and Lee 2006), distributions of aerosol
mixed-layer height (MLH, Scarino et al., 2014), and aerosol type (Burton et al., 2012).  Comparisons between HALO
and Aeolus Level 2A atmospheric optical properties products during this Aeolus Cal/Val campaign are not presented
here due to current limitations of Aeolus aerosol/cloud discrimination and low sensitivity to aerosol scattering
throughout the troposphere.  A comprehensive assessment between the Aeolus and HALO HSRL retrievals will be
carried upon the next public release of the Aeolus L2A optical properties product, which is expected in the first quarter
of 2021..

**2.3 Dropsondes**
The Yankee Environmental Systems, Inc. High-Definition Sounding System (HDSS) is an automated system
deploying the expendable digital dropsonde (XDD) designed to measure wind and pressure–temperature–humidity
(PTH) profiles, and skin sea surface temperature. A full technical description of the HDSS and XDD (referred to as
sonde hereafter) systems is provided by Black et al. (2017). HDSS XDD sonde data were used during the 2015 Polar
Winds II campaign and 2017 CPEX campaigns to validate DAWN as well as the 2015 Office of Naval Research
Tropical Cyclone Intensity (TCI) field program to study the horizontal structure of tropical cyclones (Doyle et al.
2017). The sonde measures PTH profiles at 2-Hz rate and GPS location, altitude and horizontal wind velocity at 4-
Hz, equating to 5-8 meters per vertical level.  During the Aeolus campaign, a new RH sensor deployed for the first
time within the sonde was found to have lag in response and did not have adequate sensitivity to vertical water vapor
(WV) gradients. An initial view of this is provided by Figure 14a above 5 km altitude, which will be further discussed
in Section 4. Due to this response lag, sonde WV profiles will not be discussed in detail in this paper.



A set of processing steps and filters are applied to ensure sonde data quality and that the two datasets are of
comparable vertical resolution.  The sonde wind data are first smoothed using a running 3 vertical level boxcar average
to minimize noise. Sonde wind data in the first 250 m beneath the aircraft were found to be artificially fast due to the
sonde being recently released from the aircraft, so all measurements taken within the first 250 m are removed from
any analysis. One sonde, released at 28 April at 0202 UTC, is especially noisy above 8 km so only sonde and DAWN
data below 8 km are compared for that time. In addition, the sonde altitude is adjusted by adding 40 m to each sonde
altitude measurement in order to account for a timing lag between when the sonde measurement is collected and the
time stamp of a given vertical level as suggested by M. Beaubien (personal communication). Sonde wind data within
+/- 33 m of each DAWN altitude bin are averaged, given that the DAWN pulse length projected into the vertical is
approximately 23 m.  The DAWN wind profile immediately preceding a sonde launch was used for comparison,
provided that the profile occurred within 2.5 minutes of the sonde release to minimize the impact of spatial wind
variability on the validation statistics. DAWN data met this time match criteria for 61 of the 65 sondes released across
the five flights.  Wind speed (direction) differences exceeding 10 m s$^{-1}$ (30°) were considered outliers that amounted
to 0.03% (3.57%) of 12,284 DAWN vertical bins matched with sonde data.

**306     2.4 Aeolus**

Aeolus is a direct detection Doppler wind lidar operating near 355 nm that retrieves wind speed and aerosol and cloud
profiles along a single LOS oriented 90° to the right of the Aeolus spacecraft heading (Straume et al. 2019; Kanitz et
al. 2019; Reitebuch et al. 2019 and references therein). Aeolus derives wind profiles by measuring the Doppler shift
of light backscattered from molecules (Rayleigh scattering), or clear sky aerosol and cloud particles (Mie scattering).
In this study, we analyze the Aeolus Level 2B horizontally projected LOS (HLOS) wind speed product that is
developed by the Royal Dutch Meteorological Institute (KNMI) and the European Centre for Medium-Range Weather
Forecasts (ECMWF) under ESA contract, in close cooperation with teams developing the L1B product (DLR, DoRIT)
and L2A product (Météo-France). Technical descriptions of the Aeolus wind retrieval processing and the Level 2B
product are summarized by Tan et al. (2008); Rennie et al. (2018), Witschas et al. (2020) and references therein.
The DC-8 flew along the Aeolus track for 45 to 110 minutes, flight dependent, and was along the Aeolus
track when the satellite was overhead, resulting in little spatial and temporal variability between the observations. The
evening overpass near 6 PM local time was the target for all five underflights. The Aeolus laser LOS coordinates at a



6 km altitude were used to develop the DC-8 flight track.  The DC-8 flew over a range of altitudes during the Aeolus
underpasses, from 7.5 to 12 km depending on the atmospheric conditions and the specific objectives of a given flight.

Aeolus Level 2B products provide Mie winds at 10 km intervals where clouds are present and 0.5-1.0 km

vertical spacing, and Rayleigh clear winds at near 90 km intervals and 1 km vertical spacing at altitudes sampled by
DAWN, and up to 2 km in the lower stratosphere.  When DAWN was in vector wind profiling mode, DAWN vector
winds were projected to the Aeolus viewing orientation to derive a LOS wind speed.  During the Aeolus underpass on
the first flight of the campaign (17-18 April 2019, see Table 1), DAWN was mostly operated in single LOS mode with
its beam oriented 90° to the right of the aircraft heading in order to match the sampling of Aeolus. DAWN data are
averaged to match the Aeolus horizontal and vertical bin spacing and DAWN outliers in each bin are filtered from the
averaging. At least 30 (10) valid DAWN wind retrievals must be present within a Rayleigh clear (Mie cloudy) bin to
derive a robust mean for Aeolus comparison. We used the "estimated HLOS error" parameter provided in the Aeolus
Level 2B product, where it is recommended that Rayleigh clear (Mie cloudy) winds with > 8 m/s (> 5 m/s) be excluded
(Rennie and Isaksen 2020). An DAWN-Aeolus difference exceeding 20 m/s, which only occurred in one bin, was
considered an outlier and excluded from analysis. These criteria and the duration of the Aeolus underpasses resulted
in 231 vertical Rayleigh Clear and 42 Mie Cloudy data bins distributed across 46 Aeolus Rayleigh profiles.

It is important to note that, due to a variety of technical challenges that are beyond the scope of this paper,

the Aeolus Laser-A output power gradually decreased from the time of Aeolus launch to April 2019 when the Aeolus
Cal/Val Test Flight campaign was conducted (Lux et al. 2020).  This degradation coupled with lower than expected
signal throughput in detection chain (both of which are currently being studied by the Aeolus team), served to 1)
decrease the precision of Aeolus Rayleigh and Mie derived wind products and 2) limit the ability to retrieve aerosol
products from the Mie channel under clean or very tenuous aerosol loading conditions.  ESA made the decision to
switch to Aeolus Laser-B in June 2019, which has resulted in improved laser energy output and hence improved
precision in the wind products.  In addition, anomalous signal detections have been found on the Aeolus Aeolus
Charged Coupled Device (ACCD) have been discovered (i.e. "hot pixels") and the number of affected pixels have
increased over time. A dedicated dark current calibration mode (DUDE) and an on-ground correction scheme based
on the DUDE measurements has been implemented in the ground segment in June 2019, hence has not been applied
to the measurements validated here.



Finally, comparisons of the Aeolus data quality with the ECMWF model background and collocated

CAL/VAL observations from ground-based instrumentation (including wind profilers and radiosondes) showed that
the variability of the Earth top-of-atmosphere total radiance along the Aeolus orbit cause thermal stress an
deformations of the instrument telescope which could not be fully compensated by the implemented telescope thermal
control. This has caused biases of the Aeolus L2B winds of several m/s varying along the orbit and from orbit to orbit
(Martin et al. 2020). An on-ground correction of the telescope temperature induced bias has been developed and was
implemented in April 2020. The dataset used in this comparison is hence affected by this known bias contributor.

The presented work includes preliminary data (not fully calibrated/validated and not yet publicly released)

of the Aeolus mission that is part of the ESA Earth Explorer Programme. This includes wind products from before the
public data release in May 2020 and/or aerosol and cloud products, which have not yet been publicly released. The
preliminary Aeolus wind products will be reprocessed during 2020 and 2021, which will include in particular a
significant L2B product wind bias reduction and improved L2A radiometric calibration. Aerosol, cloud, and wind
products from the April 2019 period will become publicly available by fall 2021. The processor development,
improvement and product reprocessing preparation are performed by the Aeolus DISC (Data, Innovation and Science
Cluster), which involves government and industry partners including DLR, DoRIT, ECMWF, KNMI, CNRS, S&T,
ABB and Serco, in close cooperation with the Aeolus PDGS (Payload Data Ground Segment). It is likely that
performance will improve with future reprocessing, thus extensive validation of these preliminary products will not
be emphasized in this paper.

**365    3. Flight Campaign Operations Description**

Five DC-8 flights were executed from 17-30 April 2019, four from NASA Armstrong Flight Research Center in
Palmdale, California, and one from Kona, Hawaii along flight tracks overlaid on GOES-17 imagery shown in Figure
1. Given the short duration of this campaign and other operational considerations, we generally sought to maximize
the weather targets of opportunity, with an emphasis on broad regional sampling rather than focused sampling of one
particular area or phenomenon. An exception to this was the 29-30 April flight where we transected the same regions
multiple times to study atmospheric temporal variability and instrument performance. Flight tracks were selected to
capture a diversity of wind and WV conditions while avoiding optically thick mid- to upper-level cloud layers that
can attenuate lidar signals and inhibit vertical profiling throughout the troposphere. GOES-17 satellite imagery and



NASA Global Modeling and Assimilation Office (GMAO) Goddard Earth Observing System, Version 5 (GEOS-5)
model forecasts of clouds, precipitation, and aerosols were used as guidance for flight planning.
GOES-17 visible and infrared satellite imagery was uploaded in near-real time to the DC-8, where it could
be displayed and animated during flight for situational awareness and to help identify clear or broken cloud conditions
for sonde releases. A GOES-17 Advanced Baseline Imager (ABI) Mesoscale Domain Sector (MDS) was provided by
NOAA's National Environmental Satellite, Data, and Information Service (NESDIS) along the DC-8 flight track
during almost the entirety of the 46-hour flight campaign. Images were collected every minute within an MDS, which
enable a variety of cloud process and geostationary cloud- and WV-tracked atmospheric motion vector studies that
are currently being conducted. Several examples of GOES-17 7.3 μm WV channel imagery are shown below. This
channel typically senses upwelling radiance emitted within the 500-750 hPa layer (~3 to 6 km altitude) in cloud-free
conditions, a layer that was observed during all flights.

**4. Results and Validation**
**4.1 17-18 April 2019**
The first flight of the campaign served as both a test flight to ensure that all instrumentation and components were
operating properly, as well as a science flight. The target of the flight was a strong mid-latitude cyclone over the
North Pacific, centered at 51° N, 140° W and shown in Figure 2a-b. The goal was to intersect the large dry slot (Figure
2d) of the cyclone where clear sky to broken cumulus clouds and wind speeds exceeding 50 m/s were present at flight
level at the time of the Aeolus overpass near 03 UTC (27 UTC in the cross sections).
After system testing during the first three hours of the flight, DAWN was operated in three modes specified
in Table 1. This flight was the only one to use a single LOS DAWN stare oriented 90° to the right of the aircraft
heading, which emulated the view of Aeolus and produced higher spatial resolution profiles than operations with
multiple LOS, ~0.5 km per single LOS speed profile vs 4-5 km per vector profile. During the stare periods, DAWN
was periodically reset to operate in five azimuth vector profiling mode so that DAWN wind speed and direction could
be validated with sondes being released in support of Aeolus validation. Due to the differences in DAWN operating
mode throughout the flight, all DAWN vector wind profiles were projected to the LOS horizontal wind speed along
the 90° azimuth. A time-height cross-section of DAWN and HALO data during the intensive observing period (IOP)
on April 18 is shown in Figure 3. Figure 3a shows the absolute value of the DAWN LOS wind speed to account for



changing aircraft direction. As this was the first-ever flight of HALO in the WV profiling configuration, system
testing persisted until approximately 03 UTC on 18 April.
The IOP for this flight began while the DC-8 was flying within a cold front with cirrus cloud at flight level
and opaque mid- to low-level clouds beneath, which often obscured DAWN and HALO measurements from reaching
the surface. The DC-8 was already along the Aeolus flight track at this time (white arrow in Figure 1a). After
progressing through the frontal region, a dropsonde was released at 02.58 (0236) UTC while DAWN was briefly reset
to vector wind profiling mode (Figure 4a). The DAWN, and to a lesser extent, MERRA-2 wind profile agreed with
the sonde. LOS wind speed near flight level continued to increase until a local maximum just before 0300 UTC at the
time of the Aeolus overpass at 0258 UTC. Two sonde releases were coordinated with DAWN vector profiles at ~02.83
to 03.02 (0250-0301) UTC, again demonstrating excellent agreement between sonde and DAWN (Figures 4b-c).
HALO aerosol backscatter (Figure 3c) depicted a prominent plume of elevated smoke (identified using HSRL aerosol
intensive parameters, not shown) from approximately 2 km altitude that rose to 6 km as it was advected toward the
low pressure center.
As the DC-8 approached opaque cumulus clouds close to the low center at 03.7 UTC, it turned southeastward
along a long linear segment on the return to Palmdale. Opaque mid-level cloud cover associated with the front again
inhibited profiling down to the surface, except for a few isolated profiles in breaks between clouds. While DAWN
was reset at 05.5 UTC to address the problematic INS/GPS unit, a sharp transition into a high pressure region with
weak wind flow occurred. Enhanced PBL moisture exceeding 10 g/kg near the surface was observed by HALO south
of the front around 05.5 UTC (Figure 3d). A dropsonde was released at 0605 UTC, showing winds less than 5 m/s
throughout much of the profile that were captured well by DAWN and MERRA-2 (Figure 4d), demonstrating that
DAWN can accurately measure both high and very low wind speeds. Detailed comparisons between DAWN and
MERRA-2 will be highlighted in a future paper.
Aerosol backscatter in this region was greater on average than the region near to the low pressure to the north
with a prominent dust plume in the 2-4.5 km layer after 06.25 UTC (via aerosol intensive parameters, not shown)
Between this dust and stratocumulus (stratocu hereafter) cloud layer below, an extremely dry layer was present which
could have been driven by dehydration from radiative cooling and/or subsidence. Although serving primarily as a test
flight, this first IOP provided insight into the performance of new synergistic lidar remote sensing capabilities critical
for improved understanding of atmospheric processes.




### 4.2 22-23 April 2019

The second flight on 22-23 April focused on sustained higher altitude operations, targeting a high pressure region over
the ocean where an Aeolus overpass was located near 129.5° W, followed by a low pressure region that extended
through the depth of the troposphere, centered near the Arizona – Mexico border (Figure 5a-b). Between these two
systems was a northeast-southwest oriented jet streak located across southern California and Nevada. This jet streak
had winds over 90 kts (46.3 m/s) and can be seen via dry air in GOES-17 7.3 µm imagery in Figure 1b, resulting from
subsidence within a tropopause fold. The jet streak is evident in the DAWN wind speed cross-sections during the
outbound and return legs of the Aeolus underflight at approximately 0.8 and 04.3 UTC, respectively (Figure 6a-b).
DAWN observed over 40 m/s northerly winds above 10 km within the jet core, with winds over 20 m/s extending
down to 5 km altitude. The HALO WV cross-section (Figure 6d) show a narrow filament of dry air (< 0.5 g/kg)
extending downward from 6 km at 01 UTC on 23 April to the top of the stratocu layer at approximately 01.5 UTC,
which is correlated with the jet streak observed by DAWN.
A high pressure region with forecasted weak wind flow and low total precipitable water (Figure 5d) was
sampled to the west along the triangular portion of the flight track, where a large region of low stratocu with varying
thickness and morphology were present. The aerosol distribution was complex during this timeframe with high aerosol
backscatter within the PBL overlaid by tenuous aerosol enhancements extending up to flight level (Figure 6c). These
aerosol layers exhibit a high level of correlation between the wind speed, direction, and water vapor fields and provide
insight into the complex interaction between atmospheric state, composition, and dynamics. A layer of relatively low
aerosol backscatter with some vertical variability persisted from 2 to 4 km altitude which provided insufficient signal
for DAWN wind retrieval. The flight progressed southeastward along the Aeolus track to a region of very dry air at
the southern-most edge of the line. This dry air is depicted in the GOES WV imagery in Figure 1b and can also be
seen in the HALO WV cross-section near 3 UTC, where mixing ratio below 0.1 g/kg at 4 km was observed. DAWN,
and to a lesser extent MERRA, wind profiles agreed well with sondes, as shown by the four sonde comparisons in
Figure 7.
After the Aeolus under flight near 02.5 UTC (white arrow, Figure 1b), the DC-8 proceeded northeast and
again encountered the tropopause fold which exhibited similar vertical structure as the leg three hours prior during the
outbound transit to the Aeolus underpass. The PBL depth and WVMR rose significantly after 04 UTC as the DC-8


transitioned from ocean back to land, spatial gradients of which were depicted by the 6-hour GEOS-5 forecast (Figure
5c-d).  Mountain waves of various orientations can be seen along the flight track in the contrast-enhanced GOES-17
WV image (Figure 8a) after the flight traversed southeast of Santa Catalina Island.  Complex wind, WV and aerosol
distributions were present in association with the waves within the 0-4 km layer.  A layer of high aerosol backscatter
and weak (< 5 m/s) winds was located in the lowest 0.75 km of the cross section from 04 to 04.6 (0400-0436) UTC
as the flight approached the Peninsular Range along the California coast (Figures 8a-c).  A shallow layer of ~12.5 m/s
westerly wind with reduced aerosol and drier air (Figure 8d) around 1 km altitude was beneath a filament of higher
aerosol, weaker winds, and increased moisture that extended up to 2 km.  Embedded wave structures can be seen in
the aerosol backscatter from 04.4 to 04.6 UTC.

As the flight proceeded along the US-Mexico border and then turned northward, it transected near the center

of the low pressure system (Figure 5a-b). This can be seen in the DAWN wind direction data before and after 05 UTC
in Figure bb, where wind direction changes from northwesterly (magenta) to southeasterly (red to yellow), and wind
speed reduced throughout the depth of the column.  The GEOS-5 forecast indicated that the PBL height exceeded 3.5
km across southern California and western Arizona (Figure 5c), which is consistent with HALO derived MLHs.
WVMRs  up to  7 g/kg and enhanced aerosol backscatter were observed upwards of 4 km altitude over the complex
mountainous terrain due to orographic lifting.  Deep convection had occurred earlier in the day in Arizona and had
decayed by the time the DC-8 sampled the region, leaving the remnant anvil cloud observed in the HALO aerosol
backscatter at 05.25 UTC.  The low pressure system was associated with a depression of the tropopause, which reached
a 10 km altitude.  The stratospheric layer atop this low pressure system can be seen in the HALO WV data as extremely
dry air (<.007 g/kg) from ~04.5 to 05.75 UTC, approximately 3 orders of magnitude drier than the PBL airmass below.
This transect through the stratospheric intrusion was an ideal opportunity to showcase the capability of HALO in
measuring over four orders of magnitude in WVMR from the moist PBL to the dry upper troposphere/lower
stratosphere.

After sampling the stratospheric intrusion, the flight progressed northward into Utah before turning west

across Nevada along the jet streak that was previously sampled across California.  Mountain waves were also evident
in GOES WV imagery across western Nevada near 06 UTC (Figure 1b), which can also be seen via wave structures
in layers of enhanced aerosol and water vapor measured by HALO, and DAWN wind direction variations, that
extended up to 9 km altitude.  After 06.3 UTC on the western side of the Sierra Madre, the DC-8 flew through the



Central Valley region of California while descending to land in Palmdale, where there was a notable enhancement of
aerosol backscatter in the PBL with weak wind flow, and complex vertical aerosol structures aloft.

**4.3 25-28 April 2019**

The goals of the third and fourth flights of the Aeolus campaign focused on evaluating the performance of DAWN
and HALO from the mid-latitudes to the tropics and also to the transition of wind, WV, aerosol, and cloud fields from
the sub-tropics to deep tropics. The third flight on April 25-26 took a southwesterly heading to 7° N, 133° W, then
descended as it progressed westward through the tropics at an 8 km altitude, before ascending and intersecting with
the Aeolus track along a north-northwesterly heading before landing in Kona, Hawaii (Figure 1c).  The sub-tropical
jet stream can be observed in the DAWN wind speed as the DC-8 transitioned from the mid-latitude down to the east-
west tropical line (Figure 9a).  The tropical jet extends down to approximately 6 km which correlates extremely well
with the top of elevated moist layer observed in the HALO WV profiles at approximately 22.5 UTC (Figure 9d).  As
the DC-8 proceeded westward towards the start of the Aeolus line, the mid-troposphere moist layer strengthened and
deepened, and a weak overturning circulation associated with convective detrainment could be observed in the WV
distributions, evidenced by the moist layer at and above ~8 km altitude extending northward from the deep tropics.
HALO's ability to measure and infer dynamical processes via high vertical resolution WV and aerosol measurements
is critical for improved understanding of the radiative transfer that drives large scale circulation, which can in turn
effect low tropospheric stability, cloud formation, and convective aggregation (Stevens et al 2017, Holloway et al.
2017, Mapes et al. 2017,  Lebsock et al. 2017).

Another area of interest during this flight was the long north-south transect from Palmdale to the tropics

where a variety of stratocu morphologies were present along flight track (Figure 10a), from closed cell to open cell
stratocu in the northern part of the domain (20 to 20.8 UTC), then a combination of clear sky and small closed cell
(20.8 to 21.6 UTC), and finally a larger closed cell with a different appearance than the previously observed clouds
(21.6 to 22.5 UTC).  Though wind speed was generally within the 0-10 m/s range within and above these clouds, the
wind direction, aerosol, and water vapor profiles varied significantly as seen from the DAWN and HALO data in
Figure 10.  For example, a relatively moist airmass above the PBL and a shallow PBL depth was associated with the
stratocu from 20-20.8 UTC (Figure 10e-f).  Sharp directional wind shear in the 0 to 3 km layer was present around
20.3 UTC during a brief transition to open cell stratocu (Figure 10c). Dry, high aerosol backscatter air with slightly



higher wind speed was found above the PBL from 20.8-21.6, leasing to clear sky and broken clouds. The PBL depth
and cloud top height grew with the differently textured closed cell clouds that were later encountered from 21.6 to
22.5 UTC.

Prior to 22 UTC, some thin cirrus associated with the subtropical jet stream was encountered at flight level,

which inhibited HALO profiling below 6 km. Cirrus were also encountered later in the flight around 23, 01, and 03-
05 UTC. Additionally, ice accretion on the HALO window resulted in significant near field signal and contaminated
the 532 nm cross-polarization channel. Similar near field signals were also observed on the subsequent return flight
from Kona back to Palmdale. As a result, the HSRL measurement for both of the tropical flights are limited to only
use the co-polarization channels resulting in limited retrievals of aerosol intensive parameters such as depolarization,
spectral depolarization ratio, and aerosol type. In order to increase sensitivity and enable wind retrieval in low aerosol
conditions encountered after 21.25 UTC correlated with increasing moisture aloft within a sub-tropical airmass,
DAWN was operated in a 2-azimuth, 200 pulse/azimuth mode from 22.4 to 23.6 UTC. This change helped to provide
greater vertical coverage of winds at middle levels of the cross section, though Figure 10d shows that vertical
integration at a variety of depths was still required to achieve sufficient signal for wind retrieval. A sonde was released
at 22.6 UTC where HALO observed weak aerosol signal in the 1.5 to 6 km layer. Despite this weak signal and vertical
integration required to achieve enough signal for wind retrieval, DAWN retrieved a full wind profile that agreed quite
well with the sonde (Figure 11).

As noted above, after 23 UTC, deep tropical moisture was observed by HALO, with mixing ratios exceeding

20 g/kg near the surface and exceeding 6 g/kg up to 6 km. The enhanced moisture content in the middle troposphere
could be seen in GOES WV imagery via colder temperature, indicating WV absorption from higher altitudes (Figure
1c). The tropical airmass featured weak aerosol scattering above 2 km which generally inhibited wind profiling within
the 2-6 km layer. After 02 UTC, the plane ascended as it moved northwestward to head toward the Aeolus track.
GOES WV imagery showed a sharp transition from moist to dry conditions as the aircraft crossed 10° N. This drying
can be seen in the HALO data, where mixing ratio decreased from above 5 g/kg to below 1 g/kg above 4 km. An
aerosol layer was present at 6 km which enabled wind retrieval for comparison with the Aeolus overpass in the 04-05
UTC timeframe.

The 27-28 April flight from Kona featured similar conditions to the 25-26 April flight, with tropical moisture

within and above the PBL in a relatively clean atmosphere void of significant aerosol enhancements. This flight was



designed to transect through the northern half of the ITCZ in an attempt to sample the ascending branch of the Hadley
circulation (Figure 1d). Furthermore, several diagonally-oriented transects allowed for testing and optimization of the
HALO DIAL measurements over a wide range of mid-lower tropospheric WV concentrations, demonstrating the
ability of the WV DIAL to optimize the WV absorption as a function of latitude and moisture content. Although
HALO was able to demonstrate the required spectral tuning to maintain good measurement precision over the mid-
latitude and subtropical environments, the required amount of spectral tuning required for precise measurements in
the tropics was not achieved. This was overcome by increasing the vertical averaging within the lower free troposphere
and PBL from 315 m to 585 m. The additional spectral tuning required to achieve high precision and vertical
resolution in the tropics is currently under investigation and will be implemented for future campaigns.
The DAWN and HALO wind, WV and aerosol backscatter cross-sections are shown in Figure 12. The
subtropical jet was observed by DAWN with upper level winds exceeding 30 m/s above 10 km (Figure 12a). DAWN
data show good vertical coverage despite the low aerosol loading throughout the middle free troposphere (Figure 12c).
As with the previous flight, the HALO WV and aerosol fields show a high level of correlation where the vertical
gradients of WV enhancements throughout the mid-troposphere were often correlated with gradients in aerosol
backscatter (Figure 12c-d). These gradients are likely associated with advected air masses and show the utility of
using WV and aerosol fields as atmospheric tracers for large scale motion. As with the 25-26 April flight, moistening
of the mid-troposphere was be observed as the DC-8 approached the ITCZ. The HALO WV cross-sections near the
tropics around 20 UTC (~6 N) again show evidence of overturning circulation to the higher latitudes resulting from
convective mid-level detrainment near the ITCZ.

**562     4.4 29-30 April 2019**

The final flight of the campaign sampled a similar geographic region to the oceanic portion of the 22-23 April flight,
and was focused on analyzing atmospheric spatial and temporal variability, instrument performance, and dropsonde
and HALO WV profile validation (Figure 1e). The flight began with a segment beneath 5 km extending into
southeastern California, near to regions of developing convection within an upper-level low. HALO WV mixing ratio
in this region reached 8 g/kg which provided sufficient moisture for development of deep convection (Figure 13d).
The aircraft ascended to above 9 km and crossed through the western edge of the cyclonic circulation, where a deep
layer of northerly winds exceeding 20 m/s were present (Figure 13a).



The flight progressed to a region with optically thick and spatially uniform stratocu where the aircraft
decreased altitude to near 3 km above sea level. The aircraft carried out a stair-step flight pattern at five different
altitudes over this same region (region bracketed by circles in the center of GOES imagery in Figure 1e), ascending
by approximately 1.8 km in flight level with each pass. The intent of this flight pattern was to look at the repeatability
of the lidar measurements over the same airmass and also assess the DAWN sensitivity to aerosol backscatter with
increasing flight altitude. The HALO WV and aerosol observations show very persistent, repeating patterns as the
aircraft transected the same region at different altitudes (Figure 13c-d). Extremely dry air was present just above the
stratocu tops near 2 km within a strong capping inversion (not shown), a feature which was pervasive throughout the
flight and could have been caused by a combination of radiative cooling near cloud top and subsidence, similar to the
18 April flight. The symmetry isn't quite as prominent in the DAWN data due to frequent turns, but it can be seen that
winds were continually retrieved from aircraft to stratocu cloud top until the aircraft reached 10.5 km altitude.
The flight left this stratocu region around 23.75 UTC and progressed westward to another area with clear sky
to broken clouds at 0.25 UTC (24.25 UTC on the lidar time series) to carry out lidar overpasses near an in situ spiral
location to validate the HALO WV measurements against the DLH open path measurements. As the DC-8 approached
the sampling area it was vectored by air traffic control which resulted in altitude changes around 0.3 and 0.8 UTC.
The aircraft descended down to 8 km, flew over this region, then ascended to 10 km and again flew over the same
region. A sonde was released during both transects, one corresponding to the center of the north-south leg and the
other closer to the location of a spiral near the north end of the leg. Upon a final pass near the sampling region on the
north end of the leg, the aircraft spiraled from flight altitude at ~10 km down to ~150 m above the ocean surface.
During this time the NASA LaRC DLH instrument was collecting in situ WV observations which were used to assess
the performance of the HALO and sonde derived WV profiles within the same region.
Examples of these comparison profiles between the three measurements during the descending leg of the
spiral are shown in Figure 14. Figure 14a shows the comparison between the sonde and the DLH profile. As
previously discussed, the slow response time of the humidity sensor after deployment from the DC-8 limited
meaningful observations until ~5 km above the surface. The damped response time is also evident throughout the
lower troposphere resulting in disagreement in prominent features compared to DLH. It should, however, be noted
that the moisture field within the comparison region was quite variable and some of the disagreement could result
from mismatch in sampling volumes between the two measurements. It should also be noted that only the edge of the



in-situ spiral overlapped with the multiple DC-8 remote sensing tracks and that the location of the spiral was also
offset to the northern end of the track.  Furthermore, the diameter of the in situ spiral was approximately ¼ the width
of the entire comparison track and substantial variability was observed within this volume which could explain the
high frequency variability in the DLH data around 4 km.

Figures 14b-c show the comparison between the HALO and DLH WV measurements at two different

locations along the remote sensing tracks.  These comparisons were carried out at the higher 315 m vertical resolution
as there was sufficient aerosol loading throughout the troposphere allowing for higher resolution retrievals.  For each
comparison, two independently retrieved profiles were joined using a 315 m weighted average. This was carried out
to overcome the mismatch in the sampling volumes as well as the variability in the WV field along the aircraft track
and provide a fair comparison between the two measurements.  The top portion of the profile for the comparisons in
Figure 14b-c are from 0123 UTC and extends to the highest altitude right before the start of the spiral.  In both
comparisons the top profile is used until the bottom profile is available, at which point the 315 m weighted average is
carried out.  The weighted average is applied from 8154 to 7839 m and 7090 to 6775 m for the comparisons in Figure
14b, and c, respectively. The comparison between HALO and DLH in both instances show very good agreement and
provide confidence in the validity of the measurements throughout the duration of the Aeolus campaign.  Sonde
calibration efforts are ongoing and a HALO WV validation paper is currently in development and will provide further
details on HALO performance.

After completing the descending in situ profile, the aircraft spiraled upward to 10 km and reached the Aeolus

overpass at 02.2 UTC where it stayed along the overpass track until near 03 UTC.  Very intricate structures in the
WV, aerosol, and wind fields were observed along the entire Aeolus underpass. Winds gradually accelerated to near
30 m/s along the overpass where a narrow jet streak with very low aerosol backscatter conditions was sampled near 3
UTC. This jet streak also resulted in transport of moisture from the mid-lower troposphere to the aircraft altitude (see
layer between 03-04 UTC above 8 km).  The DC-8 then proceeded directly back to Palmdale to complete the flight
campaign.

**4.6 DAWN Validation**

Figures 4, 7, and 11 show that DAWN winds agreed quite well with sonde regardless of wind speed, though

some differences are evident.  Differences should not necessarily be interpreted as "errors" because DAWN and sonde





measure winds at differing time and spatial scales, in addition to the fact that sondes drift away from the aircraft flight
track into regions not sampled by DAWN. Histograms of DAWN-sonde wind speed and direction differences are
shown in Figures 15a-b based on comparison of 61 time-matched sondes, encompassing up to 12,260 DAWN vertical
levels.  Both the wind speed and directional accuracy (e.g. bias) were minimal at 0.12 m/s and 1.02°, respectively.
Precision (i.e. root-mean-squared difference or RMSD) was 1.22 m/s and 7.6° for speed and direction, respectively.
Wind direction precision decreased with decreasing wind speed, and differences reached up to 30° for wind speed less
than 5 m/s (Figure 15c).  This is to be expected to some extent given that weak wind flows can have variable wind
direction over the typical observation/comparison periods discussed here. Wind component differences from the sonde
ranged from 1.17 (v-component, red line) to 1.29 m/s (u-component, blue line), which differed from the 2017 CPEX
campaign (Greco et al. 2020) where ~1.6 m/s RMSD for both components were found. However, sondes were released
within and near convection during CPEX 2017 where increased spatial wind variability occurs, relative to the more
quiescent conditions during the Aeolus Cal/Val campaign.  These results further reinforce the conclusion of Greco et
al. (2020), in that coherent airborne doppler wind lidar can deliver to the atmospheric research and operational
forecasting communities a low bias wind profile from 10+ km altitude with a high vertical resolution every 2-10 km
along the ground track (aircraft airspeed dependent), aerosols and clouds permitting. The combination of high
resolution, accuracy, and precision of the DAWN and HALO data, make these data extremely useful for atmospheric
and cloud process studies, and Aeolus validation.

**4.7 DAWN Comparisons with Aeolus**
Aeolus Level 2B Rayleigh clear HLOS and DAWN HLOS cross sections and profile comparisons with sonde at the
time of the Aeolus overpass from the 30 April flight are shown in Figures 16a-b.  Due in part to the issues described
in Section 2, as well as the fact that Aeolus is measuring winds from a 320 km orbit distance, the Aeolus cross section
shows much greater random variability than the DAWN cross section.  This is further depicted in the profile
comparison (Figure 16c), where a 90 km mean DAWN profile and sonde wind speed profile (projected to the Aeolus
LOS), agree extremely well but Aeolus often falls outside of the variance in each Aeolus vertical layer measured by
DAWN.  Rayleigh clear HLOS and DAWN HLOS cross sections for the other four flights are shown in Figure 17,
which again illustrate random variability in the Aeolus data relative to the more smooth and spatially-coherent DAWN
winds. A scatter diagram of Aeolus Level 2B and DAWN HLOS comparisons is shown in Figure 18, encompassing



231 vertical levels of Aeolus Rayleigh clear and 42 levels of Mie cloudy vertical bins using methods described in
Section 2.4. Aeolus Rayleigh clear had a high wind speed bias of 1.19 m/s and a RMSD of 5.14 m/s. Aeolus Mie
cloudy had a high bias of 1.98 m/s and RMSD of 4.68 m/s.  As mentioned in Section 2.4, it should be noted here that
the validated Aeolus L2B dataset is known to contain wind speed biases caused by an imperfect telescope temperature
management along the orbit and from orbit to orbit, pending the top of the atmosphere total radiance variability. This
has been shown from ECMWF model observation monitoring and from comparisons with ground-based and
radiosonde observations (Martin et al 2020). These results differ from those reported by Witschas et al. (2020) during
WindVal III and AVATARE, which could be caused by a variety of factors including differences in Aeolus laser pulse
energy and latitude, longitude, and time-dependent telescope temperature issues at the time of the campaigns, wind
conditions being sampled, sample size, and criteria used to construct the Aeolus – airborne wind lidar match database.
As mentioned previously, since the time that this Aeolus data was produced, numerous corrections to address various
instrument and data issues have been developed. Extensive validation of Aeolus is not possible here due to the
relatively small sample size of co-located data and preliminary nature of the Aeolus products. We expect that future
reprocessing of Aeolus data with improved bias correction and also greater output power from Aeolus Laser-B will
result in better data quality with improved agreement with air- and ground-based wind observations.

**5. Summary and Future Work**
This paper summarized DAWN and HALO lidar observations and the wide variety of atmospheric phenomena
sampled during the April 2019 Aeolus Cal/Val Test Flight campaign across the eastern Pacific Ocean. Though this
campaign focused on regional surveys to characterize instrument performance rather than detailed process studies,
phenomena and conditions sampled during the campaign were relatively unique for DAWN, in addition to this being
the first flight where HALO operated in WV profiling mode.  It was found that DAWN and HALO resolved complex
and detailed vertical structures and horizontal gradients associated with a variety of phenomena including mid-latitude
cyclones, jet streaks and tropopause folds, mountain waves, large scale tropical circulation, and variability associated
with changing stratocumulus cloud patterns. More focused case studies analyzing some of these features are planned
for future publication. DAWN wind retrievals generally coincided with areas of enhanced HALO aerosol backscatter
and demonstrated excellent agreement with sonde throughout the campaign. Though we were not able to validate
HALO WV profiles with sonde profiles due to sonde performance issues, validation using DLH in-situ WV data



during a spiral down to near the ocean surface indicated excellent agreement. Comparison with DLH indicated that
extremely low WVMR above stratocumulus cloud top observed by HALO during two flights was a real phenomenon,
highlighting how WV DIAL and HSRL can be used in future PBL and cloud-focused studies to resolve fine scale
features that are challenging for other passive retrieval methods. Aeolus, which had been encountering performance
issues and other technical challenges from the time of launch through when our campaign was conducted, provided
winds that differed from DAWN more than in other recent European Aeolus Cal/Val campaigns involving the DLR
2-micron coherent wind lidar. We anticipate improved agreement with DAWN when Aeolus Level 2B data is
reprocessed in the future.
This campaign provided an initial demonstration of how cloud and weather phenomena coincide with and
are modulated by variations in wind, WV, and aerosol conditions, and how such variations can be observed by airborne
lidar instruments. High precision and detailed measurements of these variables, in addition to many others such as
temperature, cloud microphysics, and precipitation profiles, are required to address key science questions posed by
the 2017 Earth Science Decadal Survey (ESAS 2017). Airborne sensors and campaigns like this Aeolus Cal/Val Test
Flight campaign are needed to collect data of sufficient precision and detail to supplement and evaluate the
performance of existing space-borne sensors.
The upcoming Convective Processes Experiment – Aerosols and Winds (CPEX-AW) campaign, scheduled
to occur in July-August 2021 and intended to operate out of Sal Island of Cabo Verde, will build upon understanding
of convective processes that has been gained from 2017 CPEX campaign datasets and models. DAWN, HALO,
dropsondes, the Airborne Precipitation and cloud Radar – 3rd Generation (APR-3), and the High Altitude Monolithic
Microwave integrated Circuit Sounding Radiometer (HAMSR, Brown et al. 2011) instruments will fly aboard the DC-
8 during CPEX-AW. CPEX-AW, in conjunction with the international Aeolus Tropical Campaign, will conduct flight
segments focused on Aeolus Cal/Val in addition to other segments to address a number of science goals including: 1)
investigating how convective systems interact with lower tropospheric and surface winds in the intertropical
convergence zone (ITCZ), 2) determining the role of aerosols, WV, winds, clouds, and precipitation and their
interactions with African weather and air quality, 3) measuring the vertical structures and variability of WV, winds,
and aerosols within the boundary layer and their coupling to convection initiation and lifecycle in the ITCZ, 4)
studying how the African easterly waves and Sahara Air Layer (dry air and dust) control the convectively suppressed
and active periods of the ITCZ. In preparation for CPEX-AW, we are continuing to improve DAWN through better



detector response, faster scanning between azimuths, and adjustment to DAWN scanning patterns which will result in
improved aerosol sensitivity, and higher spatial sampling of wind profiles and improved resolution of mesoscale wind
flows. HALO improvements for CPEX-AW include optimization of detector gain settings for improved SNR and
increasing the offset locking bandwidth of the PBL weighted transmitted wavelength to allow for optimization of the
water vapor optical depth and hence, precision within the tropical and subtropical latitudes.

**6. Acknowledgements**
We acknowledge directed funding support from the NASA Headquarters Earth Science Division that covered
execution of the flight campaign and subsequent data analysis. We thank the DC-8 team at the NASA Armstrong
Flight Research Center and the National Suborbital Education and Research Center at the University of North Dakota
for their excellent execution of the campaign. We thank the following people from NASA Langley Research Center,
Anthony Notari and David Harper for the heroic effort integrating and testing HALO prior to the campaign, Joseph
Lee for his time and effort integrating and operating HALO during the campaign, Larry Petway, John Marketon, Dave
Macdonnell, Charles Trepte, Sam Chen, Jay Yu, Abou Traore, Connor Huffine, Seth Begay, Anna Noe, Diego
Pierrottet, Alan Little, and Eric Altman for their incredible efforts in preparing and testing DAWN prior to deployment,
improving instrument performance, integrating onto the DC-8, operating throughout the campaign, and supporting
science investigations, Glenn Diskin and Joshua Digangi for providing DLH data used to validate HALO WV
observations, and Christopher Yost, Douglas Spangenberg, Thad Chee, and Louis Nguyen for providing forecasting
and flight planning support for the campaign. We also thank Sammy Henderson of Beyond Photonics for his expertise
with characterizing DAWN performance prior to integration. We thank Anne Grete Straume-Linder and Sebastian
Bley (ESA), and Oliver Reitebuch (DLR) for their contributions to description of the Aeolus mission and ALADIN
status. We thank Will McCarty and the NASA GMAO for providing GEOS-5 forecast products over the flight
campaign domain. We thank the Data Center at the University of Wisconsin-Madison Space Science Engineering
Center for providing GOES-17 imagery and NOAA NESDIS for granting 1-minute GOES-17 Mesoscale Domain
Sectors throughout the campaign. DAWN, HALO, and dropsonde datasets described in this paper can be accessed at
the NASA Langley Atmospheric Science Data Center: https://asdc.larc.nasa.gov/project/Aeolus




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



| Flight Date | DAWN Operating Mode |
|---|---|
| 17-18 April 2019 | Stare at 90 degree azimuth, 20 pulse integration (1.2 UTC to 3.7 UTC, 0.5 km/profile) <br> 5 azimuths, 20 pulses/azimuth (sporadically from 2.5 to 3.7 UTC, 4 km/profile) <br> 5 azimuths, 40 pulses/azimuth (3.7 to 7.7 UTC, 8 km/profile) |
| 22-23 April 2019 | 5 azimuths, 20 pulses/azimuth |
| 25-26 April 2019 | 2 azimuths, 200 pulses/azimuth (22.4 to 23.6 UTC, 9 km/profile) <br> 5 azimuths, 20 pulses/azimuth |
| 27-28 April 2019 | 5 azimuths, 20 pulses/azimuth |
| 29-30 April 2019 | 5 azimuths, 20 pulses/azimuth |

**Table 1: A listing of DAWN operating mode throughout the five flights of the campaign and the approximate spacing between profiles.**









**Figure 1:  DC-8 flight track (bold lines) overlaid atop GOES-17 0.64 µm visible (left) and 7.3 µm water vapor**
**brightness temperature (right, in degrees Kelvin) at (a) 00 UTC on 18 April, (b) 00 UTC on 23 April, (c) 0030**
**UTC on 26 April, (d) 2230 UTC on 27 April, and (e) 00 UTC on 30 April. DC-8 aircraft position at hourly**
**intervals is annotated on the right panels. Cyan circles indicate where dropsondes were released. White**
**arrows point to straight northwest-southeast oriented segments where the DC-8 under flew the Aeolus laser**
**track at a 6 km altitude, for durations ranging from ~45 to 110 minutes.**



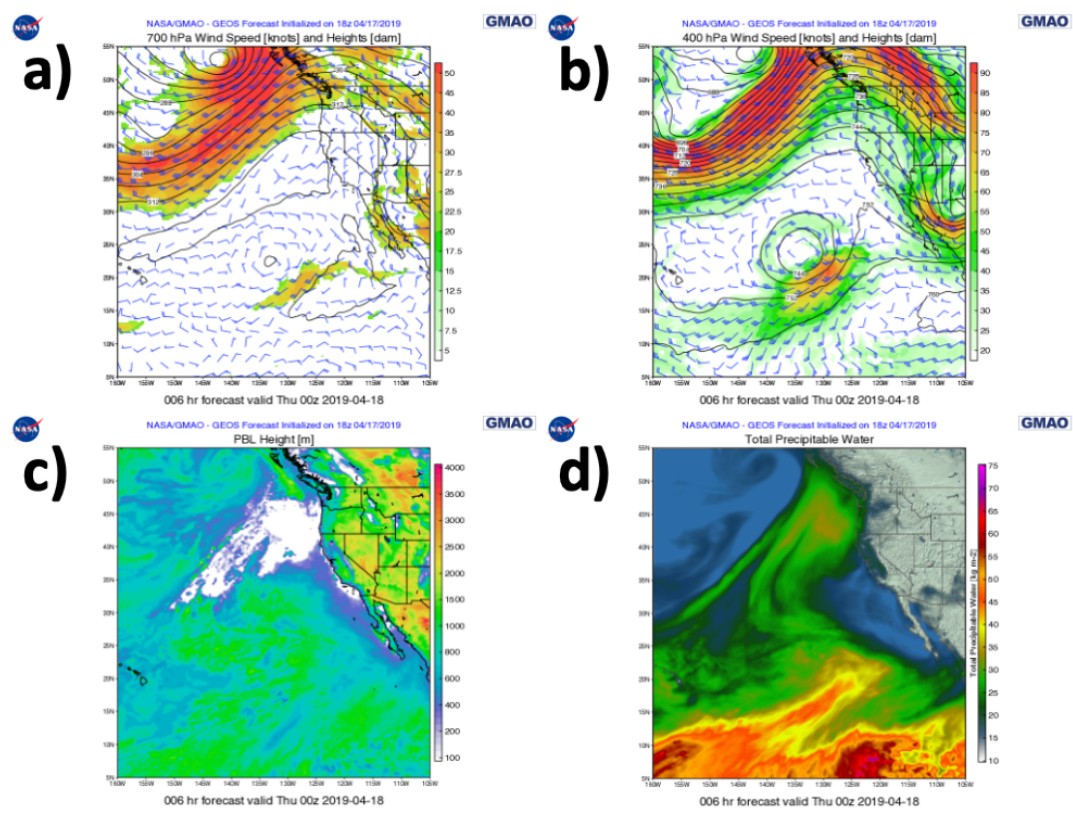

**Figure 2: NASA GMAO GEOS-5 6-hour forecast of (a) 700 hPa wind, (b) 400 hPa wind, (c) PBL height, and**
**(d) total precipitable water, valid at 00 UTC on 18 April 2019.**








**Figure 3: (a) The absolute value of the DAWN horizontal line of sight (HLOS) wind speed measurement, projected 90° to**
**the right of the aircraft heading from 0200 to 0715 UTC (i.e. 26.00-31.25 above) on 18 April 2019. The bold black line atop**
**the colored cross section indicates DC-8 flight altitude. (b) The depth of vertical signal integration required to achieve**
**sufficient signal for a DAWN wind retrieval. (c) HALO 532 nm aerosol backscatter coefficient, shown with a logarithmic**
**color scale to accentuate variations in the free troposphere aerosol distributions. (d) HALO water vapor mixing ratio also**
**shown with a logarithmic color scale. Grey areas in the DAWN and HALO cross sections indicate aircraft turns (roll ≥**
**2.5°), areas beneath opaque cloud cover, inadequate signal return inhibiting retrieval at a particular altitude, or**
**instrument downtime.**




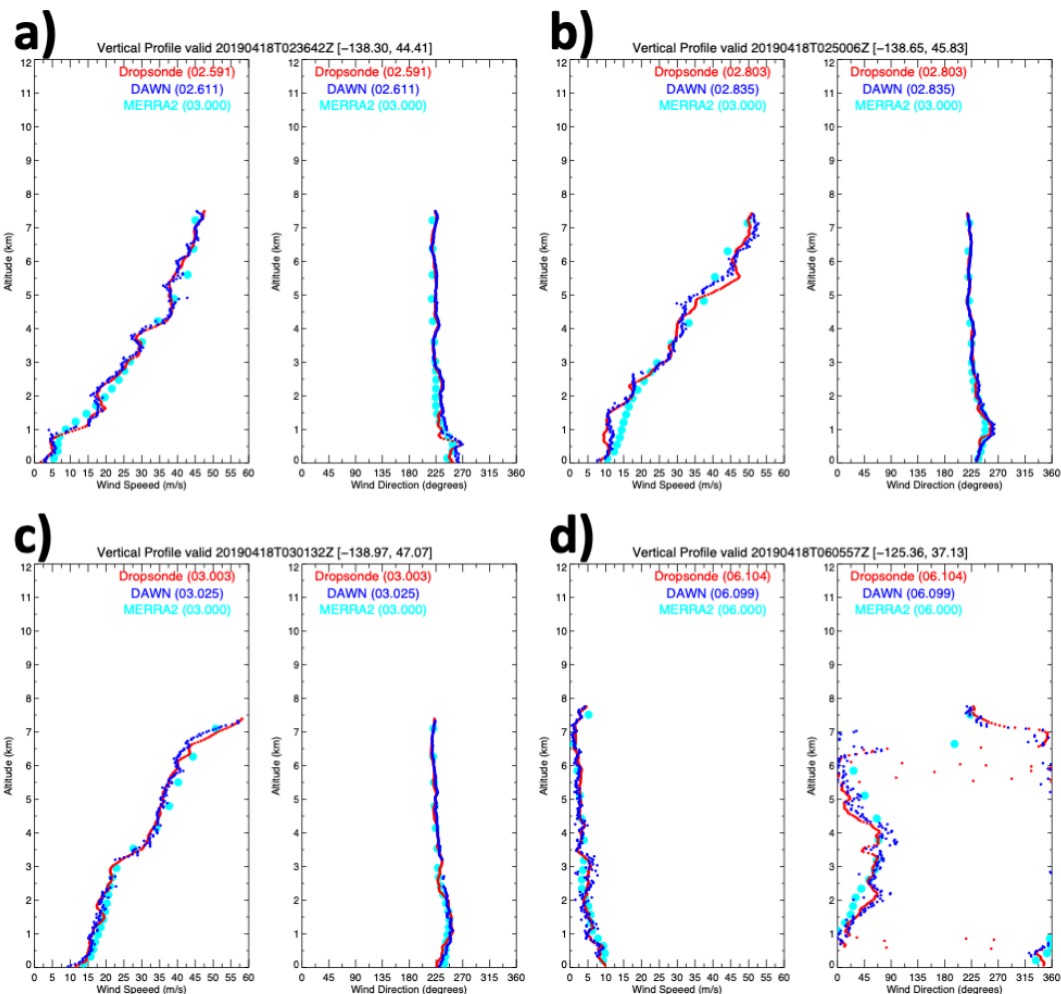


**Figure 4: Dropsonde (red), DAWN (blue), and MERRA-2 (cyan) for DAWN profiles at (a) 0236, (b) 0250, (c) 0301, and**

**(d) 0605 UTC on 18 April 2019.**








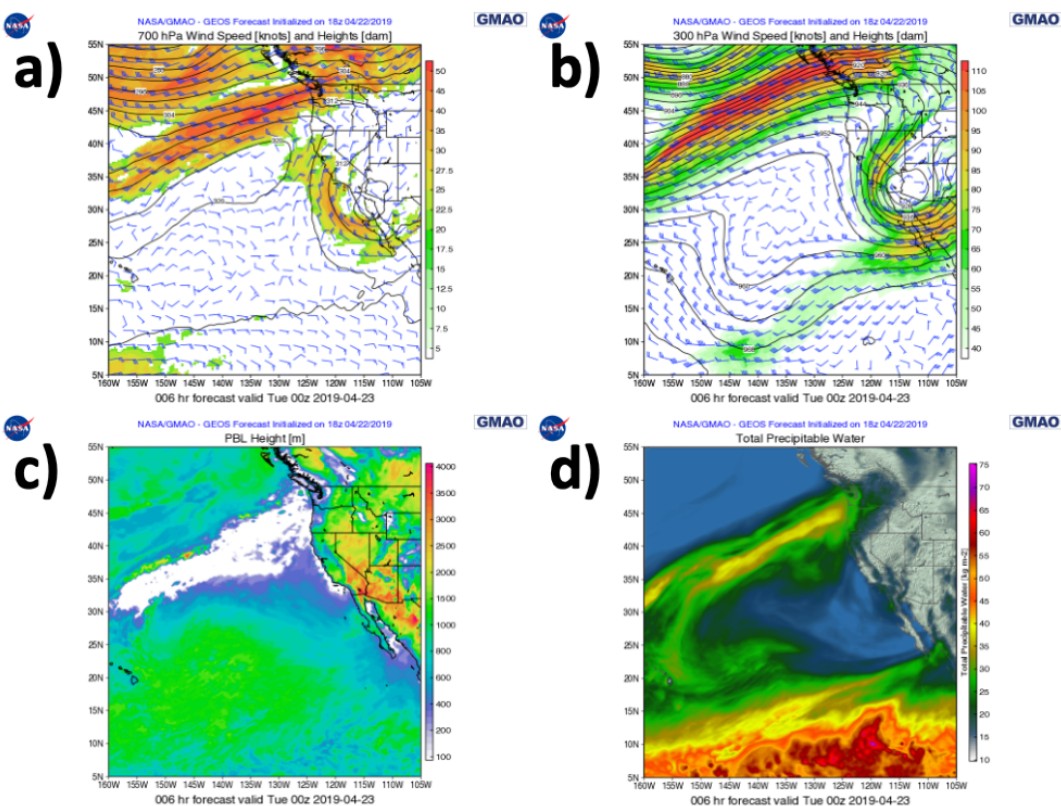

**Figure 5: NASA GMAO GEOS-5 6-hour forecast of (a) 700 hPa wind, (b) 300 hPa wind, (lower-left) PBL height, and**
**(lower-right) total precipitable water, valid at 00 UTC on 23 April 2019.**







**Figure 6: (a-b) DAWN wind speed and direction for the 22-23 April 2019 flight. © HALO 532 nm aerosol backscatter. (d)**
**HALO water vapor mixing ratio.**



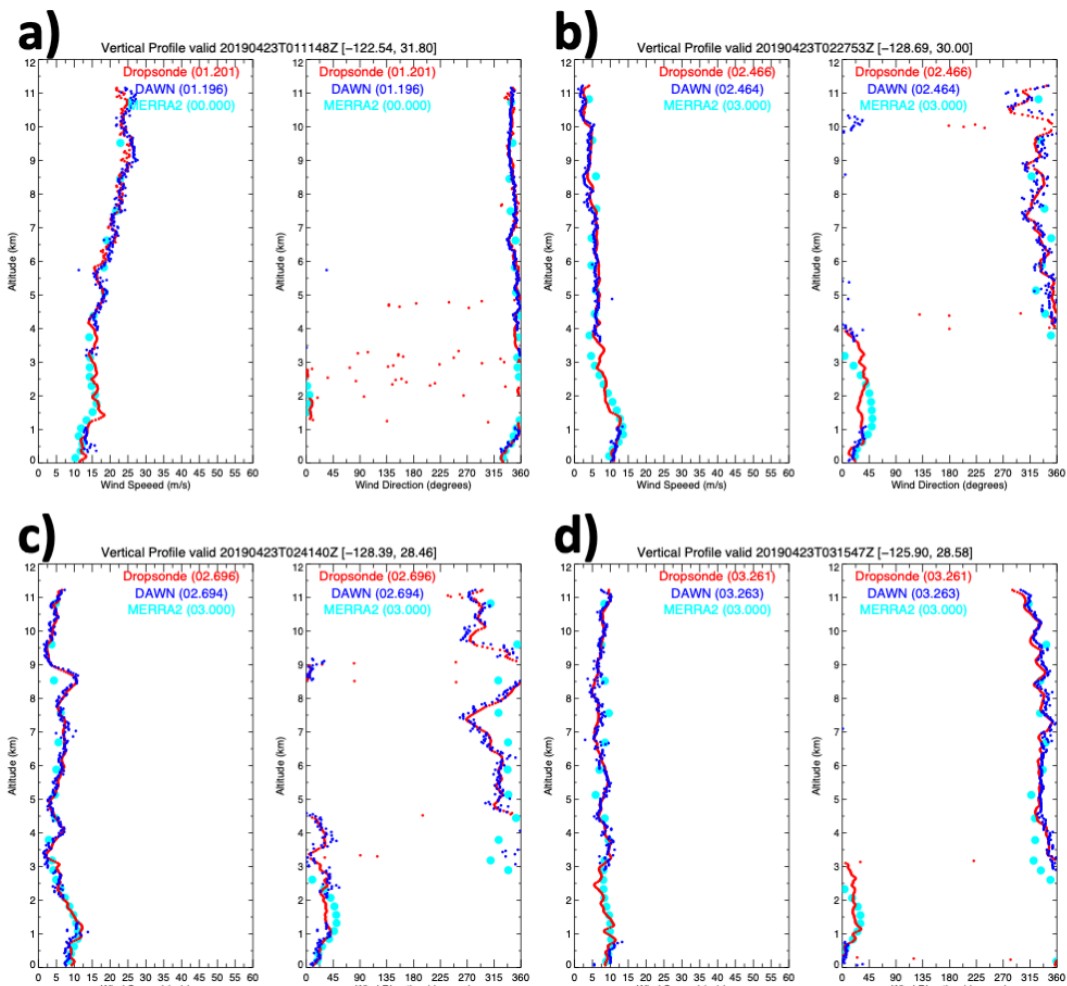

**Figure 7: Dropsonde (red), DAWN (blue), and MERRA-2 (cyan) for DAWN profiles at (a) 0111 , (b) 0227, (c) 0241, and**
**(d) 0315 UTC on 23 April 2019.**





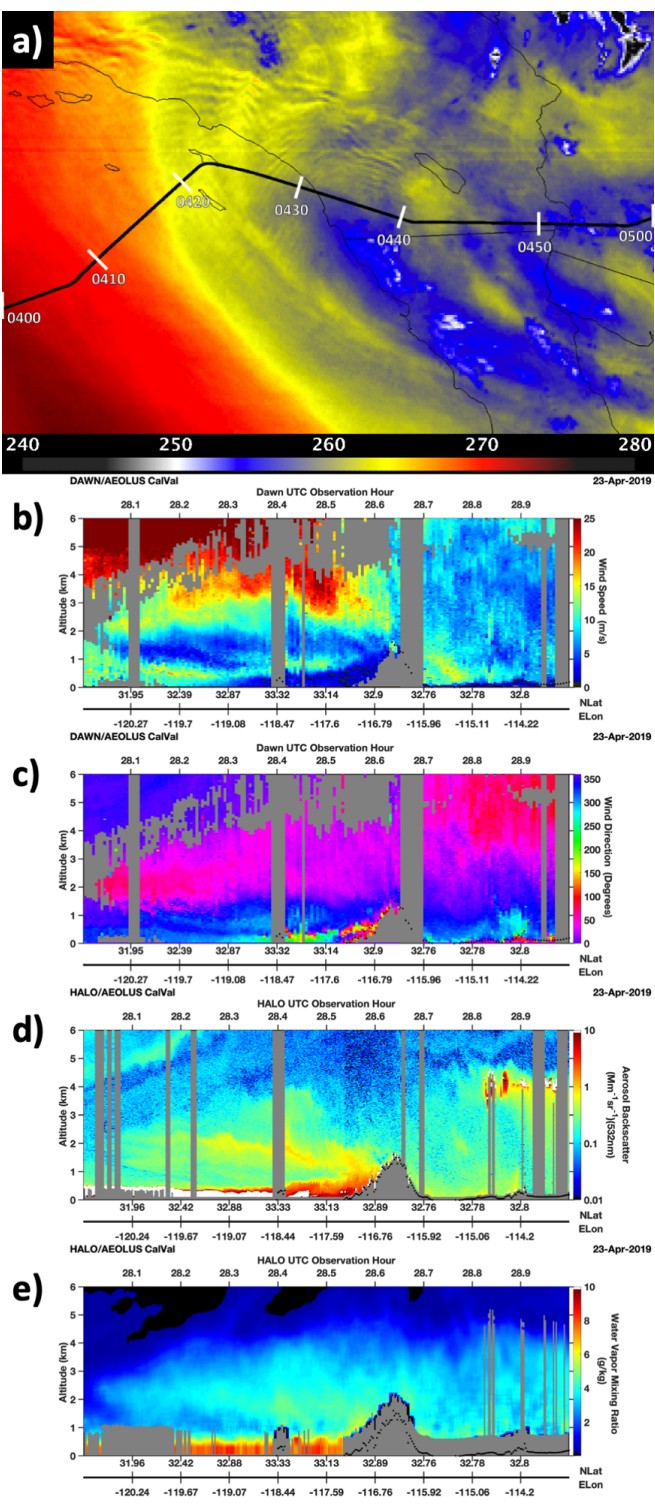




Figure 8: (a) GOES 7.3 μm water vapor brightness temperature (in degrees Kelvin) at 0426 UTC on 23 April 2019. The color scale of this image (bottom) has been compressed relative to that shown in Figure 1 to accentuate mountain wave patterns along flight track (bold black line). (b-c) DAWN wind speed and direction along the one-hour flight period shown in the GOES image. The wind speed color scale has also been compressed to accentuate wind speed variations associated with the mountain waves. (d) HALO 532 nm aerosol backscatter. (e) HALO water vapor mixing ratio, colorized with a linear scale to accentuate details.



**Figure 9: The same as Figure 6 except for the 25-26 April 2019 flight.**





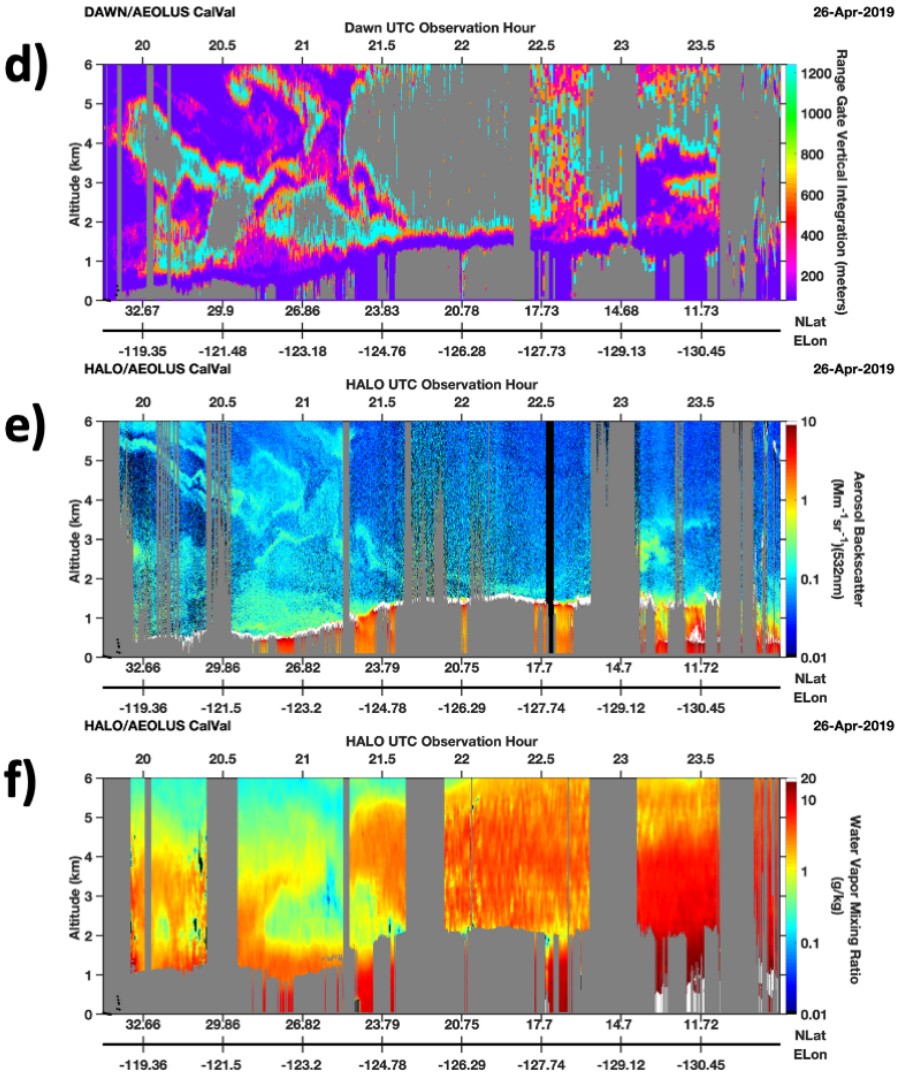

**Figure 10: (a) GOES-17 0.64 μm visible at 2200 UTC on 25 April 2019 overlaid with a nearly four-hour duration DC-8 flight track over stratocumulus and tropical convection. (b-c) DAWN wind speed and direction. DAWN wind speed color scale was compressed to 0-20 m/s to accentuate wind speed variations along flight track. Only data between 0 and 6 km altitude are shown (d) The depth of vertical signal integration required to achieve sufficient signal for a DAWN wind retrieval. (e-f) HALO 532 nm aerosol backscatter and water vapor mixing ratio.**



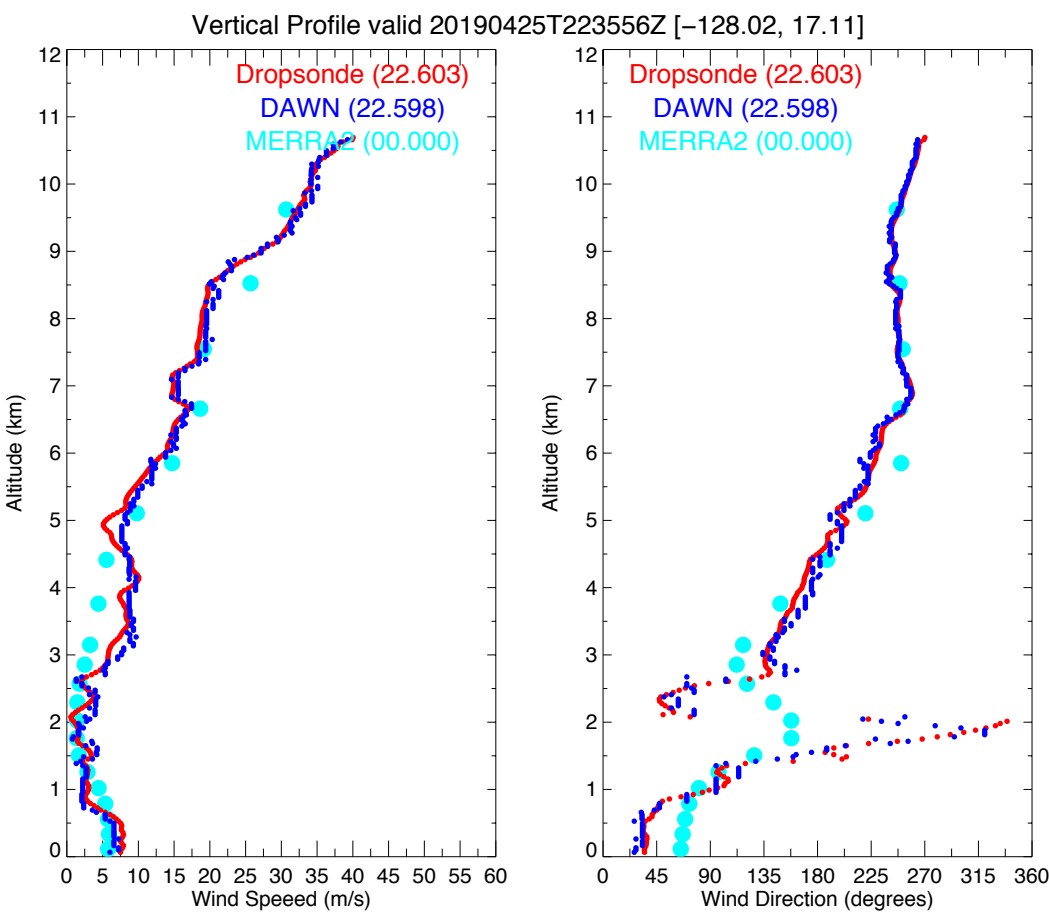


**Figure 11: Dropsonde (red), DAWN (blue), and MERRA-2 (cyan) for DAWN profiles at 2235 UTC on 25 April 2019, in**

**an airmass with weak aerosol signal within the 2-6 km altitude layer when DAWN was operating in 2-azimuth, 200**

**pulse/azimuth mode.**



**Figure 12: The same as Figure 6 except for the 27-28 April 2019 flight.**









**Figure 13: The same as Figure 6 except for the 29-30 April 2019 flight. The color scale of the DAWN wind speed is**
**compressed to 0 to 25 m/s to accentuate the lower wind speeds observed during this flight.**






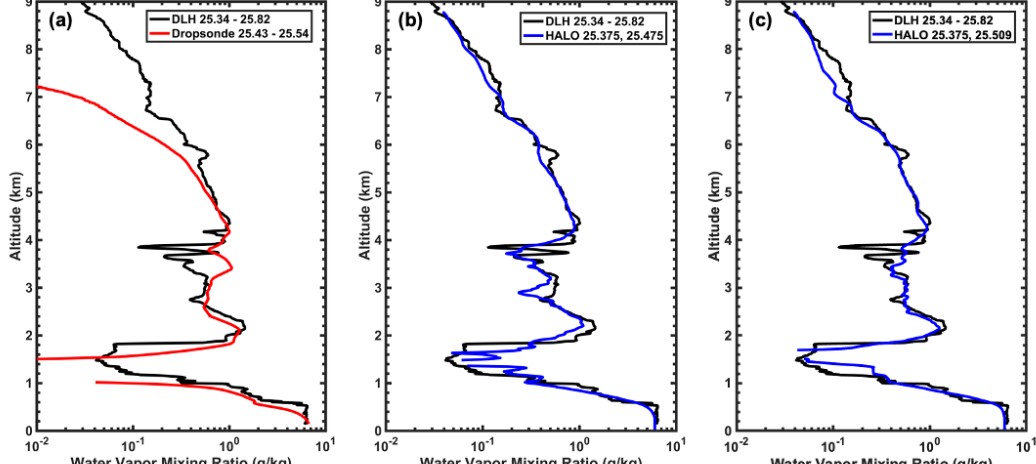

**Figure 14: HALO and dropsonde comparisons with DLH.  a) Yankee XDD dropsonde (red) comparison against DLH-**
**derived water vapor mixing ratio profile (black) from the descending DC-8 spiral.  The time of flight for the sonde and**
**those also used to generate the DLH profile are indicated in the legend.  b) HALO 315 m vertical and 12 km horizontal**
**resolution WV profile (blue) comparison against DLH.  Two HALO profiles are spliced together using the times indicated**
**in the legend to account for the heterogeneity in the WV field over the spiral location as well as to account for the spatial**
**offset between the HALO and DLH in situ spiral. c) same as b) but with a different profile chosen for the lower tropospheric**
**splice region.  Data are shown on a logarithmic scale to highlight the large dynamic range throughout the depth of the**
**profile.**



**Figure 15: A comparison of DAWN and dropsonde vector wind speed (black), and u- and v-component speed (blue and red respectively, panel a) and direction (panel b) aggregated across all vertical bins with a valid DAWN retrieval, using the methods described in Section 2.3 (panel c) DAWN-sonde wind direction difference as a function of wind speed,**



clustered into 5° and 5 m/s bins. Bins are colored by the fraction of observations within a bin relative to the total samples

in each column. The number of samples is listed within each bin.





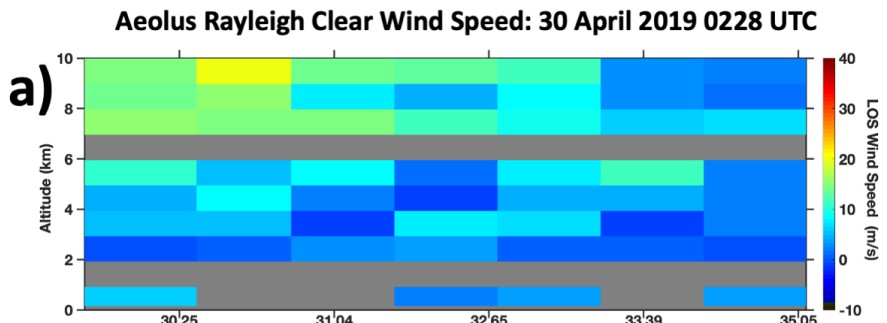

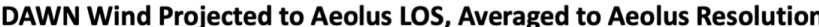

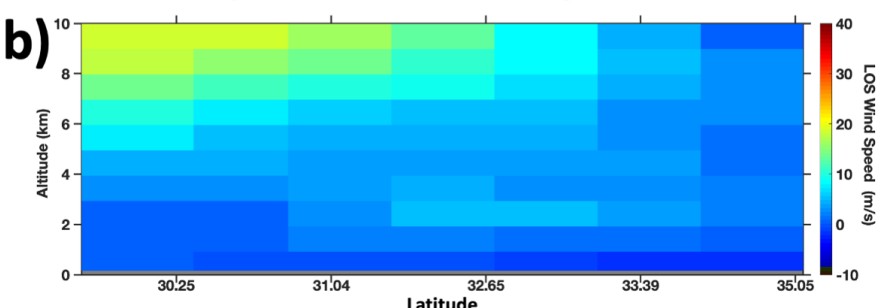

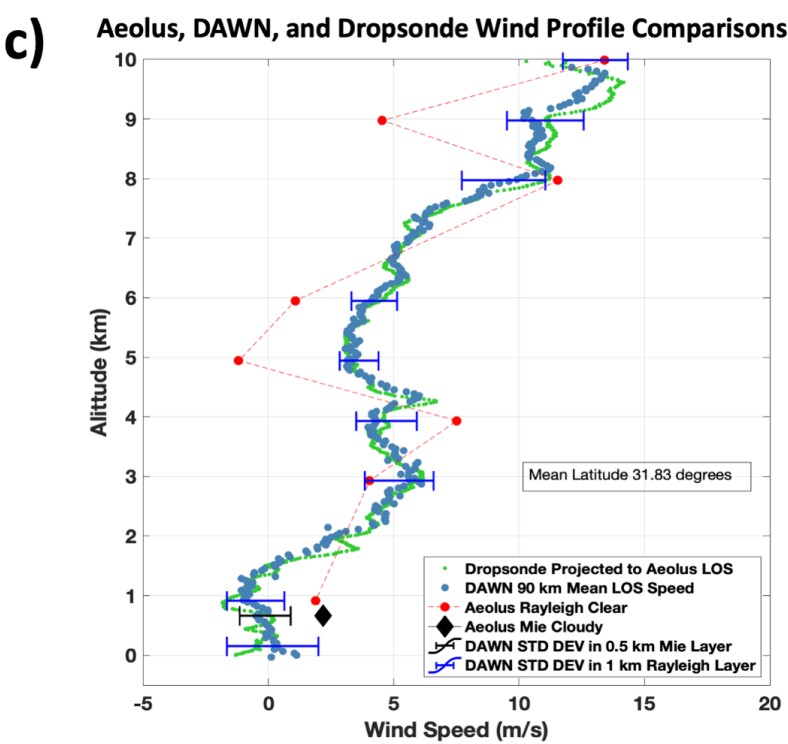



**Figure 16: a) Aeolus Rayleigh clear and b) DAWN HLOS wind speed profile cross section, coinciding with the 0228 UTC**
**Aeolus overpass on 30 April 2019. c) The mean DAWN HLOS speed aggregated across the 90-km Rayleigh clear**
**integration distance (blue), dropsonde projected LOS speed (green), and Aeolus Rayleigh Clear (red) and Mie Cloudy**
**(black diamond) speed.  DAWN variance across the Rayleigh vertical bin depth and ~90-km horizontal distance are also**
**overlaid with blue whiskers.**

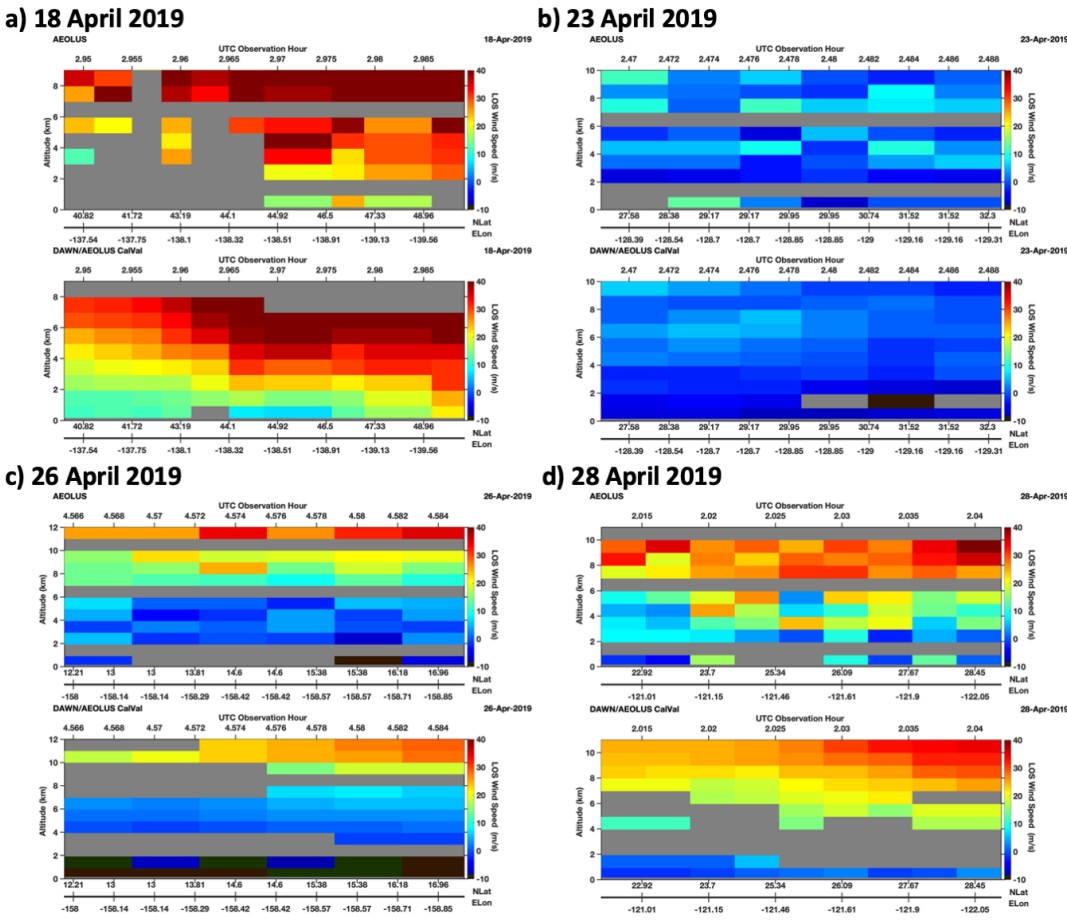

**Figure 17: Aeolus Rayleigh clear (top) and the mean DAWN HLOS speed aggregated across 90-km Rayleigh clear integration distances (bottom), analogous to Figure 16a-b, for the four remaining Aeolus under-flights.**



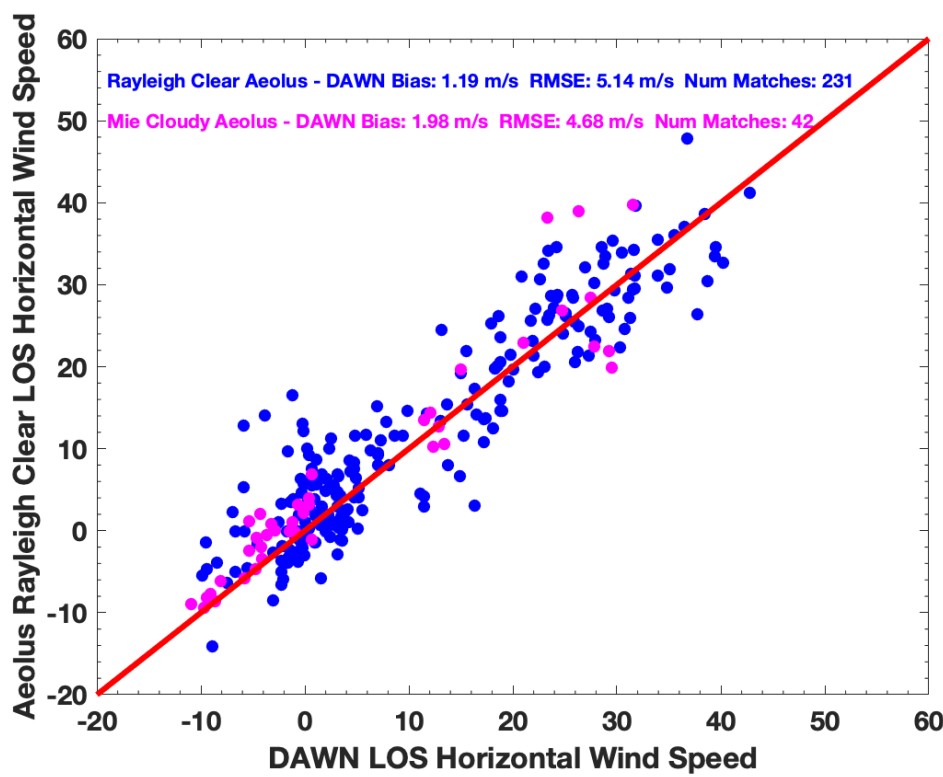


**Figure 18: Aeolus-DAWN HLOS wind comparison based on 244 Aeolus Rayleigh clear (blue) and 43 Mie cloudy**

**(magenta) vertical bins aggregated across 46 profiles.**



