# Peer review of "Airborne Lidar Observations of Wind, Water Vapor, and Aerosol Profiles During The 2 NASA Aeolus Cal/Val Test Flight Campaign"

_Atmospheric Measurement Techniques, 2020_

## Referee Comment (RC1) · Anonymous Referee #1 · 14 Dec 2020

Review of the article

**"Airborne Lidar Observations of Wind, Water Vapor, and Aerosol Profiles During The NASA Aeolus Cal/Val Test Flight Campaign"**

submitted by K. Bedka et al.
**(AMT)**

**Scientific significance: Excellent**

The paper deals with the introduction of two airborne lidar systems for measuring wind and water vapor profiles and demonstrates the possibility of using the data for calibrating or rather validating the first wind lidar in space – Aeolus. Thus, the presented results are rather significant for the lidar community but also for numerical weather forecast centers.

**Scientific quality: Good**

The paper addresses all information that are needed to understand to quality (accuracy and precision) of the measured quantities (wind and water vapor) and discusses all methodologies that are used for the data retrieval.

**Presentation quality: Excellent**

The paper manuscript is clearly structured, all used methodologies are well explained and all figures are clearly visible. Furthermore, the text is well and clearly written.

**Review Summary**

The paper manuscript "Airborne Lidar Observations of Wind, Water Vapor, and Aerosol Profiles During the NASA Aeolus Cal/Val Test Flight Campaign" by Kristopher Bedka deals with the introduction of two airborne lidar instruments by NASA that are used to measure profiles of wind and water vapor. Both instruments were flown during a recent field campaign performed in April 2019 over the Eastern Pacific Ocean aiming at demonstrating the performance of both lidar instruments but also calibrating and validating the Aeolus L2B wind products.

Both, the lidar instruments, the research flights during the campaign, and the methodologies used to retrieve and compare data are accurately explained. It is shown that the DAWN coherent wind lidar at its current state has a good performance measuring tropospheric wind profiles with almost full coverage, almost bias free (< 0.2 m/s) and with a precision of better than 1.6 m/s. The comparison of DAWN and Aeolus data from 5 Aeolus underflights is shown and discussed. It is revealed that the current Aeolus data still has a larger bias of ~ 2m/s and it is discussed that this enhanced bias may arise from thermal variations on the Aeolus telescope mirror and detector hot pixels which are corrected within a re-processed Aeolus data set.

In parallel, water vapor profiles and aerosol optical properties measured by the HALO lidar are shown and used to analyze the particular weather conditions during each of the research flights.

As already stated above, the paper manuscript is scientifically significant and well written. It is suggested to publish it after performing minor revisions as suggested below.

**Detailed comments**

- **Titel:** The title of the manuscript is "Airborne Lidar Observations of Wind, Water Vapor, and Aerosol Profiles During The NASA Aeolus Cal/Val Test Flight Campaign". Thus, the reader expects more or less only measurements that are used for Aeolus Cal/Val. However, the

comparison of DAWN and Aeolus wind data is only a very small (~5%) part of the manuscript. Neither water vapor profiles nor the aerosol optical properties data is used to compare to Aeolus. Thus, the paper mainly demonstrates what is possible with the payload flown during the campaign (also e.g. regarding boundary layer atmospheric conditions, etc.). Thus, it could be thought to make the Aeolus Cal/val less prominent within the title of the paper manuscript. However, if this was the official name of the airborne campaign it is understood that this has to be kept.

- **Introduction (general):** What is a little missing in the introduction is, what is new or rather special for the presented research results. Is it the first time that an HSRL and a DWL are flown on the same aircraft? Or is the performance of the used instruments much better than the one shown by other groups? Are there any other airborne activities to CAL/VAL Aeolus for wind and/or rather optical properties? Such kind of information would help to put the presented work in an international context.

- **Line 44:** A few acronyms, e.g. ALADIN were already introduced in the abstract. Thus, there is no need to introduce the again. The acronym ESA for the European Space Agency is not introduced, however, used in the citations. Thus, it is recommended to introduce ESA.

- **Line 48-49:** "Aeolus observes…LOS". This was already mentioned in the sentence before (at least for the wind field) → skip or harmonize these two sentences.

- **Line 66-68:** "lack hor. And vert. resolution" → It could be helpful to mention which resolutions, coverage, etc. would be required for observations to be useful for NWP (including reference).

- **Line 84-85:** Why do you use "" here? In which of the following references is this sentence written?

- **Line 115-118:** Does this mean that the beam shaping optics improve the SNR similarly than having 250 mJ pulse energy instead of 100 mJ, or just that the missing 150 mJ were partly compensated by the beam shaping optics?

- **Line 142:** Acronym INS was not introduced.

- **Line 172:** This sentence is not completely clear to me. On the one hand, quantitative numbers are missing. How good is the comparison to in-situ data? On the other hand, DAWN is expected to have no measurements at flight level, right? With a laser FWHM of 180 ns (54 m), the first 100-200 m might be influenced by the outgoing laser pulse which would shift your power spectrum towards 0 m/s. Thus, a comparison to in-situ winds at flight level might not be too meaningful. Could you clarify that?

- **Line 177-179:** It would be worth mentioning that the DLR lidar performance was determined also for airborne measurements (not from ground). So "sondes" are actually dropsondes and not radio sondes. Maybe this can be clarified in the manuscript.

- **Line 184:** "Unreliably". What does unreliably mean here? Is only the 10 Hz data not available all time, or are values corrupt (also in the 1 Hz data)?

- **Line 200:** LASE was not introduced...is it an acronym for the precursor of HALO?

- **Line 208:** "exceeded all expectations": Concerning what? Reliability? Data coverage? Accuracy? It would be good to provide a short explanation here.

- **Line 231:** water vapor → WV

- **Line 236-237:** Have you ever verified if the WV data precision is Poisson noise limited or if there are other systematic contributions to the random error?

- **Line 239:** water vapor → WV

- **Line 252:** "good agreement" What does "good" and "excellent" mean here? Can you already provide numbers that you are later discussing in section 4.4?

- **Line 262-263:** Can you quantify "favorably" here? What means "show promise" in detail?

- **Line 289:** Water vapor (WV) was introduced before

- **Line 304:** The direction seems to have more outlier than the wind speed (3.57% instead of 0.03%). Can this be related to the fact that DAWN is only measuring 5 different positions in forward direction? Would a full conical scan improve the wind direction determination?

- **Line 321-324:** Different to Aeolus, DAWN is a coherent wind lidar that "only" measures winds by analyzing the narrow-band backscatter signal from aerosols and clouds. But you use DAWN data also for the validation of Rayleigh winds that are measured in almost aerosol-free atmosphere. Can you still justify your approach?
  How do you perform the projection? How accurate do you know the viewing direction of DAWN? In case you are not pointing to the same azimuth direction, your wind measurements would need to be corrected. Did you consider this issue?

- **Line 326:** Is the laser beam pointing really 90° wrt to the heading angle or to the aircraft reference system? Do you control the angle in case the heading angle changes during the flight leg (e.g. due to changing cross wind conditions), or have you verified how constant the heading was during one Aeolus underflight legs?

- **Line 329-331:** Have you tried to use different est. error thresholds in order to verify the sensitivity to these values? This would be an interesting step as other Cal/Val teams reported at the recent Cal/Val workshop that the estimated error calculation seems to vary with time and thus different thresholds might have to be used in different periods. Thus, having the statistical comparison with different thresholds would be interesting.

- **Line 341:** two times "Aeolus", but the second "Aeolus" should be accumulation...

- **Line 360:** Maybe it would help to introduce all acronyms if not already done before...

- **Line 392:** What does UTC observation hour particularly mean? From which event is it counted? It would be better to plot UTC time in the top-x-axes in order to prevent any confusion...

- **Line 404:** With cirrus cloud(s)?

- **Line 411:** "excellent agreement" → Can you quantify? E.g., largest deviation, accuracy, precision...

- **Line 416:** "along a long" → probably correct but sounds strange and funny.

- **Line 418:** Sometimes, the UTC times are given with "." sometimes without.

- **Line 425:** Full stop missing at the end of the line.

- **Line 441:** Startocu → Stratocumulus?

- **Line 459-461:** Have you also analyzed the vertical wind speeds in the vicinity of these mountain waves? Do they lead to additional errors in the Aeolus L2B winds which do not consider vertical winds?

- **Line 514:** 21.6 UTC (UTC is missing)

- **Line 517:** was → were (?)

- **Line 528-530:** Can you give quantitative numbers here?

- **Line 611-612:** Can you give quantitative numbers here?

- **Line 631:** "reached up to 30°" → Probably you would have reached larger values, but you consider differences larger than 30° as gross outlier, right? If so, this should again be mentioned here.

- **Line 679-680:** I would also prefer to read quantitative numbers here instead of "excellent agreement".

---

## Referee Comment (RC2) · Anonymous Referee #2 · 3 Feb 2021

The paper presents the data acquired by two lidars, a Doppler wind lidar DAWN and a WV wind lidar HALO, during a two-week period in April 2019 in an effort to contribute to the Cal/Val activity of the space-borne wind lidar Aeolus of the European Space Agency. The paper aims at highlighting the DAWN and HALO measurement capabilities across a range of atmospheric conditions, and providing a comparison of DAWN measurements with Aeolus. During the camapign, HALO demonstrated the first new airborne WV DIAL capability within NASA in over 25 years. Finally, it is worth noting that the paper uses preliminary data (not fully calibrated/validated and not yet publicly released) of the Aeolus mission.

[Figure]

The paper is well written and provides the reader with a description of the HALO and DAWN systems. It demonstrates the importance of tbeing able to observe simulatneously WV, wind, aeorosol and cloud data from a single platform and a combination of remote sensing instruments.

I recommend that the paper be accepted for publication in AMT provided that a few minor comments and suggestions are accounted for.

P35 (here and in the Introduction): This would be my major comment here: I do not think that you provide any comparison of HALO measurements with Aeolus since you are not comparing aerosol/cloud related products (and you provide the reader with a good reason for that). You discuss comparison between DAWN, sondes and Aeolus, and comparison between HALO and DHL, but not comparison between HALO and Aeolus. P152: Figures 3 and 10 are the first to be mentioned in the paper..? Figure 14 is also presented before Figure (L289). The figures are not presented in the order they are numbered until Section 3. Please fix that issue. P157: GRIP acronym used before being defined here L206 Does HALO provide range-resolved CH4 measurements along the line of sight or integrated columns? At what wavelength? L226-227: what us the expected penetration depth in clouds and in the water? L232-236: are the HALO data visualized in real time in the aircraft? What do you use then to compute the dry air number density necessary for mixing ratio retrievals? L239 Can you explain how DOAD is optimized through wavelength tuning for the viewing scene? By tuning the wavelength to the side of the absorption line? Dou you use an a priori knowledge of the water content in the atmosphere to proceed with the adjustment? Is this automatized somehow? Or used induced/controlled? L264: can the use of the echo over land still be considered valid over flat terrain? Regarding the use over the ocean, is there a threshold on the wave heights beyond which the echo cannot be used to extend the WV profile? P330 do you mean error on winds > 8 m/s (5 m/s)? L341 remove one of the Aeolus L348: an -> and

---

## Author Comment (AC1) · 5 Mar 2021

**Airborne Lidar Observations of Wind, Water Vapor, and Aerosol Profiles During The NASA Aeolus Cal/Val Test Flight Campaign**
**Authors: Bedka, Nehrir, Kavaya, and Co-Authors**

**Responses To Reviewer 1 Comments**
- **Title:** The title of the manuscript is "Airborne Lidar Observations of Wind, Water Vapor, and Aerosol Profiles During The NASA Aeolus Cal/Val Test Flight Campaign". Thus, the reader expects more or less only measurements that are used for Aeolus Cal/Val. However, the comparison of DAWN and Aeolus wind data is only a very small (~5%) part of the manuscript. Neither water vapor profiles nor the aerosol optical properties data is used to compare to Aeolus. Thus, the paper mainly demonstrates what is possible with the payload flown during the campaign (also e.g. regarding boundary layer atmospheric conditions, etc.). Thus, it could be thought to make the Aeolus Cal/val less prominent within the title of the paper manuscript. However, if this was the official name of the airborne campaign it is understood that this has to be kept.

We recognize the validity of your comment, but through discussions with NASA and various presentations and publications already publicly available, the official name of the campaign that you see in the title has already been established. We had hoped to be able to include reprocessed Aeolus wind and aerosol data in this paper, which would have enabled a more comprehensive comparison with DAWN and HALO and greater presence in this paper, but this was not possible given ESA's reprocessing timeline for the Laser-A data record. Such comparisons will be done in the future after reprocessing. For the reasons here, we choose to keep the title as is.

- **Introduction (general):** What is a little missing in the introduction is, what is new or rather special for the presented research results. Is it the first time that an HSRL and a DWL are flown on the same aircraft? Or is the performance of the used instruments much better than the one shown by other groups? Are there any other airborne activities to CAL/VAL Aeolus for wind and/or rather optical properties? Such kind of information would help to put the presented work in an international context.

In terms of other activities ongoing or recently conducted at NASA regarding airborne wind or optical property profile measurements, with regards to wind, the DAWN instrument is one of NASA's airborne wind profiling instruments, but the only Doppler Wind Lidar. Other radar instruments such as APR-2 and –3 and EXRAD can retrieve vertical component winds in precipitation regions, but not horizontal winds. It was noted in the text that DAWN was flown in CPEX 2017, and previously in Polar Winds I and II. Flight opportunities are relatively infrequent at NASA, so Aeolus Cal/Val was the next flight opportunity after CPEX 2017. Aerosol profiling instrument flights focused on atmospheric composition research are quite frequent at NASA . The HSRL, HSRL-2, and more recently HALO have flown in many recent missions including SEAC4RS, ORACLES, NAAMES, ACT-America, LISTOS, FIREX-AQ, and ACTIVATE.

In terms of the novelty of these flights, this is the first time to our knowledge that a WV DIAL, HSRL, and Doppler Wind Lidar have flown together on a single aircraft. The LASE instrument flew with DAWN during the 2010 GRIP and collected relative backscatter profiles from aerosol/cloud as well as WV profiles. But this mission was one of the first for DAWN, and it did not perform as well as it did in subsequent missions, especially CPEX 2017 and Aeolus Cal/Val.

We choose not to speculate regarding how DAWN and HALO may perform better/worse relative to other international instruments of their kind.

To address your comments regarding the novelty of these flights, we have included the following sentence in the Introduction:
"To our knowledge, this is the first time that quantitative profiles of aerosol optical properties from a High Spectral Resolution Lidar (HSRL), water vapor profiles from a Differential Absorption Lidar (DIAL), and wind profiles from a Doppler wind lidar were simultaneously observed from a single aircraft"

- **Line 44:** A few acronyms, e.g. ALADIN were already introduced in the abstract. Thus, there is no need to introduce the again. The acronym ESA for the European Space Agency is not introduced, however, used in the citations. Thus, it is recommended to introduce ESA.
Done

- **Line 48-49:** "Aeolus observes…LOS". This was already mentioned in the sentence before (at least for the wind field) → skip or harmonize these two sentences.
Done

- **Line 66-68:** "lack hor. And vert. resolution" → It could be helpful to mention which resolutions, coverage, etc. would be required for observations to be useful for NWP (including reference).
The thrust of the text you comment on was directed to how current space-borne sensors do not offer the resolution nor precision required to address key process-oriented science questions, and how airborne sensors are required to help fill gaps. We have added the following text to provide an example of wind and moisture measurement requirements to address a "Most Important" ESAS (2017) PBL science question:
"For example, the ESAS (2017) Consolidated Science and Applications Traceability Matrix identifies geophysical observables and their associated accuracy required to address a number of key science questions. A ESAS (2017) "Most Important" question "*What planetary boundary layer (PBL) processes are integral to the air-surface (land, ocean and sea ice) exchanges of energy, momentum, and mass, and how do these impact weather forecasts and air quality simulations?*" requires measurement of 3-D wind vector and moisture profiles every 20 km and 3 hours, with 0.2 km vertical resolution and accuracy of 1 m/s and 0.3 g/kg, respectively, that is currently not attainable from space-borne sensors."

- **Line 84-85:** Why do you use "" here? In which of the following references is this sentence written?
The quotes were not necessary as this exact quote was not extracted verbatim from any one of these references, so they are now removed.

- **Line 115-118:** Does this mean that the beam shaping optics improve the SNR similarly than having 250 mJ pulse energy instead of 100 mJ, or just that the missing 150 mJ were partly compensated by the beam shaping optics?
We have revised the text with the following that should address your question:

"It originally generated 250 mJ per pulse using a crystal amplifier for the Genesis and Rapid Intensification Processes (GRIP) and PolarWinds I and II campaigns described below. However, this component failed and was removed. This change caused the beam size and curvature entering the beam expander (BEX) to be sub-optimum, lowering the heterodyne mixing efficiency. The amplifier space was used to locate beam shaping optics which restored the optimum beam to BEX coupling efficiency, thereby increasing signal-to-noise ratio (SNR) to a level greater than if the amplifier had been replaced."

- **Line 142:** Acronym INS was not introduced.
Done

- **Line 172:** This sentence is not completely clear to me. On the one hand, quantitative numbers are missing. How good is the comparison to in-situ data? On the other hand, DAWN is expected to have no measurements at flight level, right? With a laser FWHM of 180 ns (54 m), the first 100-200 m might be influenced by the outgoing laser pulse which would shift your power spectrum towards 0 m/s. Thus, a comparison to in-situ winds at flight level might not be too meaningful. Could you clarify that?
We are reporting on the analysis described by Greco et al. 2020. The first valid DAWN wind retrieval which as you note occurred at some distance below the aircraft was compared with the flight level wind measured in-situ by the aircraft. Though there can be some wind gradients in this 100-200 m layer you mention, the insitu data is constantly available and synchronized with every DAWN profile, serving as a stable, though imperfect reference to estimate DAWN precision at the upper portion of each profile.

- **Line 177-179:** It would be worth mentioning that the DLR lidar performance was determined also for airborne measurements (not from ground). So "sondes" are actually dropsondes and not radio sondes. Maybe this can be clarified in the manuscript.
We have clarified that the DLR comparisons were also with dropsonde.

- **Line 184:** "Unreliably". What does unreliably mean here? Is only the 10 Hz data not available all time, or are values corrupt (also in the 1 Hz data)?
We have clarified this with the revised statement "Throughout the 2019 Aeolus campaign, the INS/GPS unit attached to DAWN periodically had problems with signal acquisition that resulted in unpredictable drifts in recorded aircraft position and orientation." The 1 Hz DC-8 INS/GPS was extremely stable and was used in place of the 10 Hz unit dedicated to DAWN.

- **Line 200:** LASE was not introduced...is it an acronym for the precursor of HALO?
Lidar Atmospheric Sensing Experiment is now defined in the text

- **Line 208:** "exceeded all expectations": Concerning what? Reliability? Data coverage? Accuracy? It would be good to provide a short explanation here.

The following text was added/amended to help clarify the "exceeded all expectations statement".

Despite serving as the first set of engineering test flights, HALO exceeded all expectations regarding laser reliability, measurement sensitivity, dynamic range, and accuracy and precision (to the extent validated during this mission). The results from the Aeolus Cal/Val campaign demonstrated the first new airborne WV DIAL capability within NASA in over 25 years and provides a new observational tool to the community for future process and cal/val studies.

- **Line 231:** water vapor → WV
Corrected

- **Line 236-237:** Have you ever verified if the WV data precision is Poisson noise limited or if there are other systematic contributions to the random error?
It is difficult to assess the sources of systematic error of the HALO WV data without having a statistically significant validation data set from which to compare the lidar data against. Comparisons to the limited in situ data sets collected during this campaign and ground based radiosonde comparisons have not identified sources of systematic error and have also indicated that the measurement precision does scale with Poisson statistics to beat down shot noise and electronic noise.  We believe a detailed discussion on this subject is beyond the scope of this paper.  A validation paper is in preparation that discusses in more detail the random and systematic error of the HALO WV data products.

- **Line 239:** water vapor → WV
Corrected

- **Line 252:** "good agreement" What does "good" and "excellent" mean here? Can you already provide numbers that you are later discussing in section 4.4?
Given the very sparse sampling a detailed statistical analysis is not presented here. The following sentences were amended to present high level statistics.
"Qualitative comparisons, however, generally showed good agreement in the lower troposphere and into the PBL, where the HALO WV profiles resolve the shape and general magnitude of the WV measured by the sonde.  Comparisons with the DLH in-situ open path measurement, conducted during a spiral, showed excellent agreement with an average percent difference, above 4.5 km and below 1 km (PBL), of approximately 5%. Statistics between 1 km-4.5 km are omitted here due to the sparse sampling statistics and large variability within the in-situ sampling volume."

- **Line 262-263:** Can you quantify "favorably" here? What means "show promise" in detail?
The following text was amended/added to provide more context to the near surface comparisons.
"Preliminary results using the ocean surface echo compare favorably with the DLH in-situ observations, with an absolute difference of less than 10% from the surface up to 315 m. However, it should be noted that the lowest extent of the aircraft spiral limited the DLH observations to ~200 m above sea level so that the lidar comparisons are also limited to 200-315 m above sea level. As with the full profile comparisons to the sondes from above, the surface echo retrievals generally showed good agreement with the shape and magnitude of the near surface profiles retrieved from the sonde."

- **Line 289:** Water vapor (WV) was introduced before

**- Line 304:** The direction seems to have more outlier than the wind speed (3.57% instead of 0.03%). Can this be related to the fact that DAWN is only measuring 5 different positions in forward direction? Would a full conical scan improve the wind direction determination?

I think this is more closely tied to challenges with measuring wind direction at near zero wind speed for both the lidar and the sonde. So it is difficult to assess which wind direction measurement is truly "correct", thus excluding such points from the statistics was deemed to be a prudent decision.

**- Line 321-324:** Different to Aeolus, DAWN is a coherent wind lidar that "only" measures winds by analyzing the narrow-band backscatter signal from aerosols and clouds. But you use DAWN data also for the validation of Rayleigh winds that are measured in almost aerosol-free atmosphere. Can you still justify your approach?

Our view on validation is that in order to truly assess the accuracy of a space-borne sensor, one needs to compare with data from a reference sensor that is as close to "truth" as possible. As evidenced by experiences from DLR, NASA, NOAA, and other ground-based systems, coherent wind lidar has proven to be the most precise, and highest vertical/spatial resolution airborne wind profiling sensor that we are aware of. DLR has conducted several missions focused on Aeolus Cal/Val with their 2-micron system that are referenced in the paper text. We have done a similar activity here with DAWN. We have limited our comparisons to where sufficient signal was detectable by DAWN so we wouldn't say that the atmosphere was nearly aerosol-free. So in summary, yes we feel the approach is justified unless somehow molecules move differently than aerosols/clouds. We are not aware of a study that has investigated this aspect.

In terms of "Calibration" of Aeolus Rayleigh signals, DAWN Mie-only measurements are not useful. It is unclear to what extent (or if at all) the presence of significant aerosol would influence the Rayleigh wind retrieval. Such an analysis would require synchronized observations with a coherent and direct detection system like those collected with 2-micron and A2D by DLR.

**Lines 321-324 and 326** How do you perform the projection? How accurate do you know the viewing direction of DAWN? In case you are not pointing to the same azimuth direction, your wind measurements would need to be corrected. Did you consider this issue? Is the laser beam pointing really 90° wrt to the heading angle or to the aircraft reference system? Do you control the angle in case the heading angle changes during the flight leg (e.g. due to changing cross wind conditions), or have you verified how constant the heading was during one Aeolus underflight legs?

The reviewer brings up several good points here. We know DAWN is oriented with the 0 degree scanner position pointed at the nose of the aircraft. We know from the very precise DC-8 INS/GPS data the heading associated with every DAWN wind profile. In cross winds, the reviewer is correct that the aircraft heading required to maintain a true ground relative heading will differ, so an assumption that a 90 degree azimuth stare from DAWN would not exactly match the Aeolus pulse direction/orientation. We account for differences between aircraft heading and 90 degrees when we project vector winds to the Aeolus LOS. But during constant stare operations where we only have an LOS wind speed, we have no ability to reproject the speed to a different angle without some assumptions. We tested the sensitivity of this through assumption of a constant wind

direction profile, which is not particularly valid because there were directional changes as much as 30 degrees.  We have amended the text to discuss this point in detail.

"Though the DC-8 flew along the Aeolus laser track where it intersected the 6 km altitude, winds with some component perpendicular to the flight track (i.e. "cross winds") required the DC-8 to head into the wind to maintain a consistent heading. The difference between the DC-8 and Aeolus heading was taken into account when projecting the DAWN wind vector to the Aeolus view. During the Aeolus underpass on the first flight of the campaign (17-18 April 2019, see Table 1), DAWN was mostly operated in single LOS mode with its beam oriented 90° to the right of the aircraft heading in order to match the sampling of Aeolus. Due to strong cross winds, the DC-8 heading differed by as much as 12° from Aeolus.  Based upon intermittent vector wind profiles collected during single LOS operations (shown in Figs. 4a-c), we found that projection to a 102° orientation instead of 90° changed the LOS wind speed by up to 4 m/s.  Sensitivity tests assuming a constant wind direction profile across the entire Aeolus underpass showed that correcting the LOS wind speeds from a 90° angle to a 102° orientation resulted in a ~0.2 m/s decrease in Aeolus-DAWN Rayleigh bias but a comparable increase in RMSD. We chose not to incorporate this correction because wind direction was not truly constant throughout the 18 April underpass and the relatively negligible change in validation statistics."

- **Line 329-331:** Have you tried to use different est. error thresholds in order to verify the sensitivity to these values? This would be an interesting step as other Cal/Val teams reported at the recent Cal/Val workshop that the estimated error calculation seems to vary with time and thus different thresholds might have to be used in different periods. Thus, having the statistical comparison with different thresholds would be interesting.

We agree with you that such comparisons would be interesting if we were dealing with a reprocessed "final" Aeolus dataset.  But given the very preliminary nature of the Aeolus data from April 2019 where Laser-A output and signal throughput was far lower than at the beginning of the mission, it is unclear to what extent the expected error parameter is representative of the present day data that has better bias correction and other fixes for various Aeolus measurement/instrument issues.  So if we were to find some trends associated with expected error, would they be applicable to present day? We feel it would be best to wait for the reprocessed data to do sensitivity testing.

- **Line 341:** two times "Aeolus", but the second "Aeolus" should be accumulation…
Corrected

- **Line 360:** Maybe it would help to introduce all acronyms if not already done before...
Remaining undefined acronyms for these various Aeolus mission partners are now defined as requested

- **Line 392:** What does UTC observation hour particularly mean? From which event is it counted? It would be better to plot UTC time in the top-x-axes in order to prevent any confusion...
If a flight took off one day and persisted into the following day in terms of UTC time on the U.S. West Coast, as was the case with all flights during the campaign due to the need to underfly Aeolus which passed near California in the 02-03 UTC timeframe, we extended the UTC time axis to values beyond 24 UTC.  We recognize your concern and we considered manual

modifications to axis tick marks on all the plots and other ways to explain this. But in the end we felt that the notes in the text such as the following "wind speeds exceeding 50 m/s were present at flight level at the time of the Aeolus overpass near 03 UTC (27 UTC on the cross section time axis)." were sufficient to convey this.

- **Line 404:** With cirrus cloud(s)?
Corrected

- **Line 411:** "excellent agreement" → Can you quantify? E.g., largest deviation, accuracy, precision...
Deriving statistics for one particular profile is not especially meaningful, especially given that there were some time differential between the two observations and drift of the sonde due to high winds. But we feel that most independent observes would agree that the two observations are quite close to each other. But the term "excellent" is in the eye of the beholder, so we have revised the text to say "strong correlation": "Two sonde releases were coordinated with DAWN vector profiles at ~02.83 to 03.02 (0250-0301) UTC demonstrating strong correlation between sonde and DAWN (Figures 4b-c)."

- **Line 416:** "along a long" → probably correct but sounds strange and funny.
Replaced along with for

- **Line 418:** Sometimes, the UTC times are given with "." sometimes without.
Including a decimal point depends on the precision required to identify a given feature

- **Line 425:** Full stop missing at the end of the line.
Corrected

- **Line 441:** Startocu → Stratocumulus?
Line ~425 noted that we shortened stratocumulus to stratocu at that point and thereafter

- **Line 459-461:** Have you also analyzed the vertical wind speeds in the vicinity of these mountain waves? Do they lead to additional errors in the Aeolus L2B winds which do not consider vertical winds?
We do not collect vertical motion measurements with DAWN, just vertical profiles of the horizontal winds derived from off-nadir laser pulses

- **Line 514:** 21.6 UTC (UTC is missing)
Corrected

- **Line 517:** was → were (?)
Corrected

- **Line 528-530:** Can you give quantitative numbers here?
Per the comment above, we feel that most observers would agree with our assessment that there is strong agreement between the DAWN profile and sonde, and that reporting statistics on each and every radiosonde comparison in the paper that we show would lessen the readability of an

already relatively long and dense text.  We do recognize your concern about the use of terms such as "excellent" and "quite well", in that quite and excellent are subjective.  We have removed the word quite from this sentence, leaving us with "DAWN retrieved a full wind profile that agreed well with a sonde"

**- Line 611-612:** Can you give quantitative numbers here?
We agree with you, however, deriving statistics for one particular profile is not especially meaningful, especially given that there was substantial time and volume differential between the two observations.  The following text was amended/added:

"As discussed above, the comparisons with the DLH in-situ open path measurement conducted during a spiral showed good agreement with an average percent difference above 4.5 km and below 1 km (PBL) of approximately 5%. Statistics between 1 km-4.5 km are omitted here due to the sparse sampling statistics and large variability within the in-situ sampling volume.  The limited comparison between HALO and DLH show very good agreement and provide confidence in the validity of the measurements throughout the duration of the Aeolus campaign. A HALO WV validation paper is currently in preparation and will provide further details on HALO performance with independent in-situ and space-based observations."

**- Line 631:** "reached up to 30°" → Probably you would have reached larger values, but you consider differences larger than 30° as gross outlier, right? If so, this should again be mentioned here.
We have revised this portion of the text to reflect uncertainty with wind direction measurement from DAWN and sonde at very low wind speed.
"Wind direction precision decreased with decreasing wind speed and was lowest for wind speed less than 5 m/s (Figure 15c).  This is to be expected given that weak wind flows can have variable wind direction over the typical observation/comparison periods discussed here. For example, the sonde wind direction  profile deviated from DAWN quite significantly in the 6-7 km altitude layer in Figure 4d where wind speeds less than 2.5 m/s were measured. It is unclear to what extent the sonde can precisely measure wind direction at very slow wind speed, so we feel that use of a 30° gross outlier filter is justified."

**- Line 679-680:** I would also prefer to read quantitative numbers here instead of "excellent agreement".
The RMSD values for DAWN have been inserted and the term "excellent agreement" was replaced with "close agreement"

---

## Author Comment (AC2)

**Airborne Lidar Observations of Wind, Water Vapor, and Aerosol Profiles During The NASA Aeolus Cal/Val Test Flight Campaign**

**Authors: Bedka, Nehrir, Kavaya, and Co-Authors**

**Responses To Reviewer 2 Comments**

P35 (here and in the Introduction): This would be my major comment here: I do not think that you provide any comparison of HALO measurements with Aeolus since you are not comparing aerosol/cloud related products (and you provide the reader with a good reason for that). You discuss comparison between DAWN, sondes and Aeolus, and comparison between HALO and DHL, but not comparison between HALO and Aeolus.

The text below was in the original manuscript and discusses the reasoning for not showing the HALO/Aeolus aerosol comparisons. The correlation of the aerosol profiles measured between the two instruments during all of the overpasses was very low and we did not feel it was appropriate to include those results given the preliminary nature of the Aeolus L2A optical properties.

"Comparisons between HALO and Aeolus Level 2A atmospheric optical properties products during this Aeolus Cal/Val campaign are not presented here due to current limitations of Aeolus aerosol/cloud discrimination and low sensitivity to aerosol scattering throughout the troposphere. A comprehensive assessment between the Aeolus and HALO HSRL retrievals will be carried upon the next public release of the Aeolus L2A optical properties product, which is expected in the first quarter of 2021."

P152: Figures 3 and 10 are the first to be mentioned in the paper..? Figure 14 is also presented before Figure (L289). The figures are not presented in the order they are numbered until Section 3. Please fix that issue.

We recognize that typically the first Figure cited is assigned Figure 1. But in our case, we briefly refer ahead to make a short point related to wind retrieval vertical coverage and issues with the dropsonde important to the methodology. Given the flow of the narrative, where we progress sequentially through the various flights in chronological order, it would not make sense to introduce these Figure panels ahead of all the others, so we choose to keep the Figure numbering as is.

P157: GRIP acronym used be- fore being defined here

This has been corrected.

L206 Does HALO provide range-resolved CH4 measurements along the line of sight or integrated columns? At what wavelength?

The following text was amended/added:

"Though HALO has successfully flown in several field campaigns in the $CH_4$ DIAL/HSRL configuration, providing weighted $CH_4$ columns at 1645 nm in addition to aerosol/cloud profiling, the 2019 Aeolus Cal/Val campaign was the maiden deployment for the $H_2O$ DIAL/HSRL configuration."

L226-227: what us the expected penetration depth in clouds and in the water?

The penetration depth in cloud and water is dependent on the cloud/ocean extinction in addition and the instrument noise floor. Cloud extinction is highly variable. Penetration depths can be as large as 3 optical depths which can be as deep as 40-50 m in non-productive open ocean waters. We believe this kind of information is beyond the scope of the paper to warrant inclusion of this point in the text.

L232-236: are the HALO data visualized in real time in the aircraft? What do you use then to compute the dry air number density necessary for mixing ratio retrievals?

The following text was added: "The HALO WV data are calculated in real-time for instrument and flight sampling optimization using a standard atmosphere model to convert the measured DAOD to mass mixing ratio."

L239 Can you explain how DOAD is optimized through wavelength tuning for the viewing scene? By tuning the wavelength to the side of the absorption line? Do you use an a priori knowledge of the water content in the atmosphere to proceed with the adjustment? Is this automatized somehow? Or used induced/controlled?

The reviewer is correct, we tune the transmitted wavelength to the side of the line to either increase or decrease absorption based on the water vapor concentration. Although there is a theoretical basis for optimizing the precision of the WV retrieval through optimization of the measured DAOD (Remsberg and Gordley 1978), in practice, the instrument noise characteristics come into play. The optimization is currently done manually by assessing simultaneously a combination of the measured DAOD and the received signal SNR. No a priori information is required. We are in the process of automating this optimization routine. Given this is a high-level mission overview paper we feel that this information is beyond the scope of the paper. We have added the following text for clarification:

"For the Aeolus Cal/Val campaign, HALO was able to demonstrate a precision of better than 10% with 6 km along track averaging when the WV differential absorption optical depth (DAOD) was optimized by tuning the wavelength along the side of the absorption line for the specific viewing scene."

L264: can the use of the echo over land still be considered valid over flat terrain? Regarding the use over the ocean, is there a threshold on the wave heights beyond which the echo cannot be used to extend the WV profile?

The text in the manuscript was incorrect.  The on/off sampling time is 1 ms which corresponds to ~20 cm translation between the on and off footprints at the surface.  The surface height and albedo variations within the laser footprint over this scale is negligible.  We have demonstrated surface echo DIAL measurements on the HALO CH4 configuration, and those results are currently in preparation for publication.  The following text was added to clarify the lack of land surface echo retrievals in this paper.

"Given the majority of the campaign was over the ocean, the detector gain settings were not optimized to keep the land surface echo on scale and therefore the surface echo retrievals are not employed over land for this study."

P330 do you mean error on winds > 8 m/s (5 m/s)?

The word winds is included in the existing text, so no edits were made to address this question

We used the "estimated HLOS error" parameter provided in the Aeolus Level 2B product, where it is recommended that Rayleigh clear (Mie cloudy) winds with > 8 m/s (> 5 m/s) be excluded (Rennie and Isaksen 2020).

L341 remove one of the Aeolus

This has been corrected

L348: an -> and

This has been corrected